# UTS: When monotonic value factorisation meets non-monotonic and stochastic targets

## Abstract

In the paradigm of centralised training with decentralised execution, monotonic value decomposition is one of the most popular methods to guarantee consistency between centralised and decentralised policies. This method always underestimates the value of the optimal joint action and converges to the suboptimal because it can only represent values in the restricted monotonic space. A possible way to rectify this issue is to introduce a weighting function to prioritise the real optimal joint action and learn biased joint action-value functions. However, there may not exist an appropriate weight to solve more general tasks with non-monotonic and stochastic target joint action-values. To solve this problem, we propose a novel value factorisation method named uncertainty-based target shaping (UTS), which projects the original target to the space that monotonic value factorisation can represent based on its stochasticity. First, we employ networks to predict the reward and the embedding of the next state, where the prediction error quantifies the stochasticity. Then, we introduce a target shaping function to replace the targets for deterministic suboptimal with the best per-agent value. Since we remain the optimal policy unchanged during shaping, monotonic value decomposition can converge to the real optimal with any original targets. Theoretical and empirical results demonstrate the improved performance of UTS in the task with non-monotonic and stochastic target action-value functions.

## 1 Introduction

Recent progress in cooperative multi-agent reinforcement learning (MARL) has shown attractive prospects for various real-world applications, such as the smart grid management (Aladdin et al., 2020) and autonomous vehicles (Zhou et al., 2021). Due to practical communication constraints and intractably large joint action space, decentralised policies are often used in MARL. It is possible to use extra information from the environment and other agents in a simulated or laboratory setting. Exploiting this information can significantly benefit policy optimisation and improve learning performance (Foerster et al., 2016; 2018; Rashid et al., 2020).

In the paradigm of centralised training with decentralised execution (CTDE), agents' policies are trained with access to global information in a centralised way and executed only based on local histories in a decentralised way (Oliehoek et al., 2008; Kraemer & Banerjee, 2016). One of the most significant challenges is to guarantee the consistency between the individual policies and the centralised policy, i.e., the Individual-Global Max (IGM) principle (Son et al., 2019). In value decomposition methods, QMIX (Rashid et al., 2018) applies a monotonic mixing network to factorise the joint $Q$-value function, which naturally meets the IGM principle. Inspired by QMIX, many algorithms are proposed to improve coordination from different perspectives, e.g., multi-agent exploration (Mahajan et al., 2019), role-based learning (Wang et al., 2020b;c), and policy-based algorithms (Wang et al., 2020d). However, they can represent the same class of joint $Q$-values as QMIX because they use the same monotonic mixing network.

However, since QMIX can only represent values in the restricted monotonic space, there exists a gap between the approximated joint $Q$-values and the non-monotonic target values $Q$ from the environment. In some special tasks, QMIX can underestimate the value of the real optimal joint action and converge to a suboptimal (Son et al., 2019; Mahajan et al., 2019; Rashid et al., 2020). Recent works try to solve this representational limitation from two different perspectives. This first category

introduces the joint actions (Wang et al., 2020a; Mahajan et al., 2021) or pairwise interactions (Böhmer et al., 2020; Li et al., 2021) into centralised learning to achieve full representational capacity for target $Q$-values. However, learning such centralised values is difficult due to the large joint action space. Another category is to prioritise the real optimal joint action and learn biased joint $Q$-value functions. WQMIX (Rashid et al., 2020) introduces a weighting function into the projection from the target value functions to the joint $Q$-values and uses it to down-weight every suboptimal action whose target value is less than the current estimate. However, the poor empirical results on decentralised micromanagement tasks in StarCraft II show that it is difficult to apply the weighting function to more general tasks (Rashid et al., 2020; Wang et al., 2020a). We prove that the weight for the suboptimal should be small to help QMIX focus on the representation of the optimal joint $Q$-value and solves the non-monotonic targets. In addition, the weight for each action should be uniform to avoid overestimating the suboptimal whose target is large with a low probability. Due to this contradiction, there may not exist an appropriate weight to recover the optimal policy when the target is non-monotonic and stochastic.

This paper aims to take a step towards the latter category. We propose a novel value factorisation method named uncertainty-based target shaping (UTS), which projects the original target to the space that monotonic value factorisation can represent based on its stochasticity. First, we formulate two prediction problems and use the prediction error to quantify the stochasticity of the target joint $Q$-values. We employ a reward predictor and a state predictor to approximate the standard deviation of the reward and the embedding of the following state. The predicted standard deviation of the reward and the error of the state are expected to be significant if the pair leads to stochastic reward and stochastic state transition, respectively. Then, we introduce a shaping function to project the original targets $Q$ to the monotonic space and keep the optimal policy unchanged. In practice, the best action value network is applied to predict the action value when each agent gets the cooperation of others. We use the minimal best per-agent values over all agents to replace the suboptimal target. We prove that this shaping can guarantee that all shaped targets are tractable for monotonic value decomposition. In addition, the optimal policy is the same for the original and the shaped targets. Therefore, QMIX can achieve full representational capacity for the shaped target $Q$-values rather than the original ones and converge to the real optimal.

We list our main contributions as follows:

- We first analyse the limitations of the weighting function in value decomposition methods and show that it cannot guarantee convergence to the optimal when target $Q$-value functions are non-monotonic and stochastic.

- We introduce a target shaping function to project the original targets to a monotonic space, which ensures that QMIX can converge to the optimal with any original targets.

- We propose uncertainty-based target shaping and empirically show its improved performance in practice, especially in tasks with non-monotonic and stochastic targets.

## 2 BACKGROUND

A fully cooperative multi-agent task in the partially observable setting can be formulated as a Decentralised Partially Observable Markov Decision Process (Dec-POMDP) (Oliehoek & Amato, 2016), consisting of a tuple $G = \langle A, S, \Omega, O, U, P, R, n, \gamma \rangle$, where $a \in A \equiv \{1, \ldots, n\}$ describes the set of agents, $S$ denotes the set of states, $\Omega$ denotes the set of joint observations, and $R$ denotes the set of rewards. At each time step, an agent obtains its observation $o \in \Omega$ based on the observation function $O(s, a) : S \times A \to \Omega$, and an action-observation history $\tau_a \in T \equiv (\Omega \times U)^*$. Each agent $a$ chooses an action $u_a \in U$ by a stochastic policy $\pi_a(u_a|\tau_a) : T \times U \to [0, 1]$, forming a joint action $\mathbf{u} \in \mathbf{U}$, which leads to a transition on the environment through the transition function $P(s', r|s, \mathbf{u}) : S \times \mathbf{U} \times S \times R \to [0, 1]$, where $r \in R$ is the team reward. The goal of the task is to find the joint policy $\pi$ which can maximise the joint $Q$-value function $Q^\pi(s_t, \mathbf{u}_t) = \mathbb{E}_{s_{t+1:\infty}, \mathbf{u}_{t+1:\infty}}[G_t|s_t, \mathbf{u}]$, where $G_t = \sum_{i=0}^\infty \gamma^i r_{t+i}$ is the discounted return.

VDN (Sunehag et al., 2018), QMIX (Rashid et al., 2018), and WQMIX (Rashid et al., 2020) are $Q$-learning algorithms for the fully cooperative multi-agent tasks, which estimate the joint $Q$-value function $Q(s, \mathbf{u})$ as $Q_{tot}$ with specific forms. Considering only a fully-observable setting for ease of representation. VDN factorises $Q_{tot}$ into a sum of individual $Q$-value functions. By contrast,

QMIX applies a state-dependent monotonic mixing network to combine per-agent $Q$-value functions with the joint $Q$-value function. The restricted spaces of all $Q_{tot}$ that linear and monotonic value factorisations can represent are:

$$\mathcal{Q}^{lvf} := \{Q_{tot}|Q_{tot}(s,\mathbf{u}) = \sum_{a=1}^{n} Q_a(s,u_a), Q_a(s,u) \in \mathbb{R}\} \tag{1}$$

$$\mathcal{Q}^{mvf} := \{Q_{tot}|Q_{tot}(s,\mathbf{u}) = f_s(Q_1(s,u_1),...,Q_n(s,u_n)), \frac{\partial f_s}{\partial Q_a} \geqslant 0, Q_a(s,u) \in \mathbb{R}\}, \tag{2}$$

where the monotonic mixing function $f_s$ is parametrised as a feedforward network, whose non-negative weights are generated by hypernetworks that take the state as input.

WQMIX views QMIX as an operator $\Pi_{\text{Qmix}}Q := \arg\min_{Q_{tot} \in \mathcal{Q}^{mvf}} \sum_{\mathbf{u} \in \mathbf{U}} (y(s,\mathbf{u}) - Q_{tot}(s,\mathbf{u}))^2$, which can be separated into two parts: the first computes the $Q$-learning target $y(s,\mathbf{u}) := \mathbb{E}[r + \gamma \max_{\mathbf{u}'} Q_{tot}(s',\mathbf{u}')]$, and the second projects the target into $\mathcal{Q}^{mvf}$. Then, WQMIX introduces the weighting function into the projection to prioritise the optimal joint $Q$-value:

$$\Pi_{\text{WQmix}}Q := \arg\min_{Q_{tot} \in \mathcal{Q}^{mvf}} \sum_{\mathbf{u} \in \mathbf{U}} w(s,\mathbf{u})(y(s,\mathbf{u}) - Q_{tot}(s,\mathbf{u}))^2 \tag{3}$$

Since it is computationally infeasible to obtain the optimal joint $Q$-value, WQMIX proposes Centrally-Weighted QMIX (CW-QMIX) and Optimistically-Weighted QMIX (OW-QMIX) to place more importance on better joint actions. In practice, the weighting function $w : S \times \mathbf{U} \to (0,1]$ is:

$$w^{cw}(s,\mathbf{u}) = \begin{cases} 1 & y(s,\mathbf{u}) > \hat{Q}^*(s,\hat{\mathbf{u}}^*) \text{ or } \mathbf{u} = \hat{\mathbf{u}}^* \\ \alpha & \text{otherwise} \end{cases}, w^{ow}(s,\mathbf{u}) = \begin{cases} 1 & y(s,\mathbf{u}) > Q_{tot}(s,\mathbf{u}) \\ \alpha & \text{otherwise} \end{cases}, \tag{4}$$

where $\hat{\mathbf{u}}^* = \arg\max_{\mathbf{u}} Q_{tot}(s,\mathbf{u})$ is the current greedy joint action, $\hat{Q}^*(s,\hat{\mathbf{u}}^*)$ is an approximation of the true optimal value function which is not constrained to be monotonic.

## 3 CASE STUDIES

In this section, we examine the weightings of WQMIX when the $Q$-learning targets are non-monotonic and stochastic. Since the non-linear mixing network in WQMIX makes many analysis methods inapplicable, we use WVDN to analyse the fundamental limitations of the weighting. The WVDN operator is defined as $\Pi_{\text{WVDN}}Q := \arg\min_{Q_{tot} \in \mathcal{Q}^{lvf}} \sum_{\mathbf{u} \in \mathbf{U}} w^{ow}(s,\mathbf{u})(y(s,\mathbf{u}) - Q_{tot}(s,\mathbf{u}))^2$, where $w^{ow}(s,\mathbf{u})$ is the optimistic weighting from (4).

### 3.1 NON-MONOTONIC TARGETS

QMIX cannot represent joint $Q$-value functions that are characterised as non-monotonic. Although this representational limitation will not necessarily result in the convergence to the suboptimal, QMIX fails to recover the optimal policy with some specific non-monotonic targets (Son et al., 2019). To accurately represent the optimal joint $Q$-value rather than the suboptimal ones, Rashid et al. (2020) introduce the weighting function in (4) to down-weight every suboptimal action. However, the choice of weight $\alpha$ is crucial to ensure that WQMIX can overcome the limitations of QMIX.

**The weight $\alpha$ should be small enough to prioritise estimating the real optimal action value.** We start by analysing a matrix game in Figure 1a, where each joint action receives a team reward. Our study focuses on the greatest value $\alpha$ to converge to the optimal when the policy is initialised to be suboptimal. Suppose the training dataset has a $\epsilon$-greedy distribution, where the greedy action is $(u_1^2, u_2^2)$. We find that $\alpha$ for WVDN should be smaller than $\frac{3\epsilon(a-c)+\epsilon^2(c-b)}{(\epsilon-3)^2(c-b)}$ to recover the optimal joint policy. The proof can be found in Appendix A.2. In Figure 1b, We show that $\alpha$ for WQMIX also should be smaller than a specific value to overcome the non-monotonic targets. The complete results are provided in Appendix B.

### 3.2 STOCHASTIC TARGETS

The stochasticity of the target value functions comes from both rewards and transitions, i.e., $P(s',r|s,\mathbf{u})$ is not deterministic. Traditional $Q$-learning algorithms solve this problem by calculating

the weighted sum of possible rewards and values of the next state multiplied by their probabilities, i.e., $Q(s, \mathbf{u}) = \sum_{s'} \sum_r [P(s', r|s, \mathbf{u})r + \gamma P(s', r|s, \mathbf{u}) \max_{\mathbf{u}'} Q(s', \mathbf{u}')]$. By contrast, the weighting function in WQMIX prioritises actions whose targets are greater than the estimated joint $Q$-value from the current mixing network. This weighting can be seen as a type of distribution shaping, leading to potential risks in policy learning.

**The weight $\alpha$ should be large to anchor down the overestimated value for the suboptimal whose target is huge with a small probability**. Since $r(s, \mathbf{u})$ and $\max_{\mathbf{u}'} Q(s', \mathbf{u}')$ play an equal role in the target value function, we take the matrix game with stochastic rewards as an example. To get a better understanding of the effect of the weight $\alpha$ in the stochastic environment, we show a numerical example in Figure 1c, where the suboptimal action $\mathbf{u}^s = (u_1^2, u_2^2)$ receives 12 with probability 0.5 and 0 with 0.5, where the optimal is $\mathbf{u}^* = (u_1^1, u_2^1)$. Suppose the training dataset is fixed, where the data has a $\epsilon$-greedy distribution and the greedy action is $\mathbf{u}^s$. Let $Q(\mathbf{u})$ denote the true joint $Q$-value. Since this matrix game only involves one state, we omit the state $s$ in the value function for simplicity.

| $U_2$\\$U_1$ | $u_1^1$ | $u_1^2$ | $u_1^3$ |
|---|---|---|---|
| $u_2^1$ | a | b | b |
| $u_2^2$ | b | c | c |
| $u_2^3$ | b | c | c |

(a) A non-monotonic game.

| Algo \ $\epsilon$ | 1.0 | 0.5 | 0.25 |
|---|---|---|---|
| Analysis | 3.0 | 15.0 | 52.6 |
| WVDN | 10.5 | 42.2 | 100.5 |
| OW-QMIX | 12.2 | 43.2 | 103.2 |
| CW-QMIX | 11.5 | 29.6 | 83.5 |

(b) Minimal $1/\alpha$.

| $U_2$\\$U_1$ | $u_1^1$ | $u_1^2$ | $u_1^3$ |
|---|---|---|---|
| $u_2^1$ | 8 | 6 | 6 |
| $u_2^2$ | 6 | 12/0 | 6 |
| $u_2^3$ | 6 | 6 | 6 |

(c) A stochastic game.

| $U_2$\\$U_1$ | $u_1^1$ | $u_1^2$ | $u_1^3$ |
|---|---|---|---|
| $u_2^1$ | 8 | 6 | 6 |
| $u_2^2$ | 6 | 6 | 6 |
| $u_2^3$ | 6 | 6 | 6 |

(d) QMIX ($\alpha$=1).

| $U_2$\\$U_1$ | $u_1^1$ | $u_1^2$ | $u_1^3$ |
|---|---|---|---|
| $u_2^1$ | 7.34 | 7.29 | 6.06 |
| $u_2^2$ | 7.29 | 7.23 | 6.05 |
| $u_2^3$ | 6.05 | 6.03 | 5.92 |

(e) OW-QMIX($\alpha$=0.5).

| $U_2$\\$U_1$ | $u_1^1$ | $u_1^2$ | $u_1^3$ |
|---|---|---|---|
| $u_2^1$ | 7.33 | 7.25 | 6.33 |
| $u_2^2$ | 7.25 | 7.2 | 6.28 |
| $u_2^3$ | 6.33 | 6.28 | 5.45 |

(f) CW-QMIX($\alpha$=0.5).

| $U_2$\\$U_1$ | $u_1^1$ | $u_1^2$ | $u_1^3$ |
|---|---|---|---|
| $u_2^1$ | 7.29 | 7.67 | 6.42 |
| $u_2^2$ | 7.67 | 10.9 | 6.62 |
| $u_2^3$ | 6.43 | 6.63 | 5.73 |

(g) OW-QMIX ($\alpha$=0.1).

| $U_2$\\$U_1$ | $u_1^1$ | $u_1^2$ | $u_1^3$ |
|---|---|---|---|
| $u_2^1$ | 7.98 | 7.98 | 6.33 |
| $u_2^2$ | 7.98 | 7.98 | 6.33 |
| $u_2^3$ | 6.33 | 6.33 | 5.51 |

(h) CW-QMIX ($\alpha$=0.1).

Figure 1: (a) The payoff matrix for a one-step non-monotonic game, where $a > c > b$. (b) The minimal $1/\alpha$ to recover the optimal policy with different $\epsilon$ in the non-monotonic game in (a), where $a = 8, b = -12, c = 6$. (c) The payoff matrix for a one-step stochastic game. (d-h) The estimated joint $Q$-values $Q_{tot}$ returned from QMIX and WQMIX. Boldface means the optimal joint action from the payoff matrix or the greedy joint action from the $Q_{tot}$.

Figure 1d-h demonstrate the joint $Q$-values $Q_{tot}(\mathbf{u})$ returned from QMIX and WQMIX with different $\alpha$. QMIX is a special case of WQMIX ($\alpha = 1$) and thus represents the joint $Q$-value perfectly. By contrast, WQMIX with a small weight is drawn to the suboptimal $\mathbf{u}^s$ and thus has an incorrect argmax. OW-QMIX returns an overestimated $Q_{tot}(\mathbf{u}^s) = 10.9 > Q_{tot}(\mathbf{u}^*)$ and thus converge to the suboptimal $\hat{\mathbf{u}}^* = \mathbf{u}^s$. In particular, CW-QMIX places more importance on $\mathbf{u}^s$ when it is not the greedy action and receives the reward of 12, leading to an overestimated value $Q_{tot}(\mathbf{u}^s) > Q(\mathbf{u}^s) = 6$. With the increase of $Q_{tot}(\mathbf{u}^s)$, $\mathbf{u}^s$ becomes the greedy action once $Q_{tot}(\mathbf{u}^s) > Q_{tot}(\mathbf{u}^*)$. In this situation, CW-QMIX does not place the $\alpha$ for $\mathbf{u}^s$ when it receives the reward of 0, leading to the decrease of $Q_{tot}(\mathbf{u}^s)$. As a result, CW-QMIX is stuck in this loop and cannot converge to any policy.

**Theorem 1** *Let* $\Pi_{\text{WVDN}}Q := \arg\min_{Q_{tot} \in \mathcal{Q}^{lvf}} \sum_{\mathbf{u} \in \mathbf{U}} w^{ow}(s, \mathbf{u})(Q(s, \mathbf{u}) - Q_{tot}(s, \mathbf{u}))^2$ *and* $w^{ow}(s, \mathbf{u})$ *is the optimistic weighting from (4). Then* $\exists Q$ *such that* $\arg\max \Pi_{\text{WVDN}}Q \neq \arg\max Q$ *for any* $\alpha \in (0, 1]$.

The proof is provided in Appendix A.2. An intuitive understanding is that the weight $\alpha$ in WVDN has an upper bound to deal with non-monotonic $Q$-learning targets and has a lower bound to anchor down the overestimated suboptimal value due to undesirable weights in the stochastic environment. The empirical results in Appendix B show that WQMIX also suffers from this contradiction and fails to converge to the optimal.

## 4 METHOD

In this section, we introduce a shaping function to generate the shaped target $Q^f(s, \mathbf{u})$ to replace the original target $Q(s, \mathbf{u})$. The shaping function has the following properties: 1) **policy invariance**, which means the optimal policy remains unchanged during shaping, i.e., $\arg\max_{\mathbf{u}} Q^f(s, \mathbf{u}) = \arg\max_{\mathbf{u}} Q(s, \mathbf{u})$. 2) **full representational capacity**, i.e., all shaped joint $Q$-values should belong

to a subset of $\mathcal{Q}^{mvf}$. Note that we achieve the full representational capacity w.r.t the shaped targets rather than the original ones. QMIX can recover the optimal joint policy with arbitrary original targets based on these properties. To this end, we first predict the standard deviation of the reward and the embedding of the next state. We then split the stochastic joint $Q$-values from the sampled data according to the prediction error. Next, we employ a network that estimates the best action value for each agent. Based on the prediction error and the best action values, we use the shaping function to generate shaped target joint $Q$-values, which can be represented by monotonic value decomposition.

## 4.1 UNCERTAINTY ESTIMATION

The stochasticity of the target joint $Q$-value comes from the stochastic transition of the reward and the state, i.e., $P(s', r|s, \mathbf{u})$, which can be viewed as a type of the aleatoric uncertainty (Der Kiureghian & Ditlevsen, 2009). We propose a solution to this undesirable stochasticity using two prediction problems, where the output is the standard deviation of the reward and the embedding of the following state of a given state-action pair. This involves three neural networks: a reward predictor network, a state predictor, and a fixed and randomly initialised target network. The reward predictor network takes the state and joint action to the reward $\hat{r}(s, \mathbf{u}) : S \times U \to \mathbb{R}$ and the uncertainty, i.e., the standard deviation $\hat{\sigma}(s, \mathbf{u}) : S \times U \to \mathbb{R}$. Based on the Bayesian deep learning method (Kendall & Gal, 2017), the reward predictor network is trained by minimising the following loss:

$$\sum_{b=1}^{B}(\frac{1}{2\hat{\sigma}^b(s, \mathbf{u})^2}(r^b(s, \mathbf{u}) - \hat{r}^b(s, \mathbf{u}))^2 + \frac{1}{2}\log \hat{\sigma}^b(s, \mathbf{u})^2), \tag{5}$$

where $B$ denotes the batch size of transitions sampled from the replay buffer. The standard deviation is expected to be higher if the state-action pair leads to stochastic rewards.

Inspired by RND (Burda et al., 2018), the target network for the second prediction problem takes a state to an $m$-dimensional embedding $g(s) : S \to \mathbb{R}^m$, where $g(s)$ is a fixed and randomly initialised network. The state predictor $h(s, \mathbf{u}; \theta_h) : S \times U \to \mathbb{R}^m$ is trained to minimise the prediction error $E(s, \mathbf{u}, s') = \|g(s') - h(s, u)\|^2$ with respect to its parameters $\theta_h$, where $s'$ denotes the next state in the sampled trajectory. The prediction error is expected to be higher if the state-action pair results in stochastic transitions. Combining the reward and the state predictors, we define an indicator function $T(s, \mathbf{u}) = \mathbf{1}_{(\hat{\sigma}(\mathbf{s}, \mathbf{u}) > \mathcal{D} \bigcup E(\mathbf{s}, \mathbf{u}, \mathbf{s}') > \mathcal{E})}$, where $\mathcal{D}$ and $\mathcal{E}$ are hyper-parameters.

## 4.2 SHAPING FUNCTION

QMIX cannot represent the non-monotonic target $Q$-value because it imposes the monotonic constraint on the relationship between the joint $Q$-value and the individual $Q$-values. Therefore, we can represent the suboptimality of uncooperative actions rather than their exact values. To quantify the suboptimality of actions, we define the best action value function $q_a(s, \tau_a, u_a) = \max_{\mathbf{u}_{-a}} Q(s, u_a, \mathbf{u}_{-a})$ as the greatest $Q$-value of $u_a$ when others perform optimally to cooperate with it. $q_a(s, \tau_a, u_a)$ is less than the optimal $Q$ value if $u_a$ is not part of the real optimal. Inspired by BAIL (Chen et al., 2020), we approximate $q_a(s, \tau_a, u_a)$ by minimising the following loss:

$$\sum_{b=1}^{B}\sum_{a=1}^{n} K(s, \tau_a, u_a)[q_a(s, \tau_a, u_a) - y_a^b(s, u_a)]^2 + \lambda\|\theta_{q_a}\|, \tag{6}$$

where $B$ denotes the batch size of transitions sampled from the replay buffer, $y_a^b(s, u_a) = \hat{r}(s, \mathbf{u}) + \gamma \max_{u_a'} q_a(s', \tau_a', u_a')$ is the target, $\lambda$ is the regularisation coefficients, $K(s, \tau_a, u_a) = 1$ for the $Q$-value whose target is stochastic or smaller than the current estimate, i.e., $y_a^b(s, u_a) < q_a(s, \tau_a, u_a) \bigcup T(s, \mathbf{u}) = 1$, and $K(s, \tau_a, u_a) = k \gg 1$ otherwise. Equipped with $q_a(s, \tau_a, u_a)$, we can compute the original target $y(s, \boldsymbol{\tau}, \mathbf{u})$ for joint $Q$-value $Q_{tot}$ and projects it into $y^f(s, \boldsymbol{\tau}, \mathbf{u})$. We define the corresponding shaping function as follows:

$$y^f(s, \boldsymbol{\tau}, \mathbf{u}) = \begin{cases} \min_{a \in \mathbb{A}} q_a(s, \tau_a, u_a) & T(s, \mathbf{u}) = 0 \text{ and } y(s, \boldsymbol{\tau}, \mathbf{u}) \leqslant Q_{tot}(s, \boldsymbol{\tau}, \hat{\mathbf{u}}^*) \\ y(s, \boldsymbol{\tau}, \mathbf{u}) & \text{otherwise} \end{cases}, \tag{7}$$

where $Q_{tot}(s, \boldsymbol{\tau}, \mathbf{u}; \theta) = f_s(Q_1(\tau_1, u_1), ..., Q_n(\tau_n, u_n))$ is the joint $Q$-values parametrized by $\theta$, $\frac{\partial f_s}{\partial Q_a} \geqslant 0$, $y(s, \boldsymbol{\tau}, \mathbf{u}) = r(s, \mathbf{u}) + \gamma \max_{\mathbf{u}'} Q_{tot}(s', \boldsymbol{\tau}', \mathbf{u}'; \theta')$, $\hat{\mathbf{u}}^* = \arg\max_{\mathbf{u}} Q_{tot}(s, \mathbf{u})$, and $\theta'$ denotes the parameters of a target network that are periodically copied from $\theta$.

Then, we introduce a shaping function $f(s, \mathbf{u})$ to project the target $Q(s, \boldsymbol{\tau}, \mathbf{u})$ into the restricted monotonic space that QMIX can represent. In practice, we replace the target $Q(s, \boldsymbol{\tau}, \mathbf{u})$ by the minimal value of $q_a(s, \tau_a, u_a)$ for deterministic and suboptimal joint $Q$-values. The goal is to ensure that all shaped target value functions at a given state belong to the restricted monotonic space and that the target for the optimal is greater than the suboptimal. We do not change the stochastic targets because QMIX can deal with them already. To improve learning efficiency, we also remain the target unchanged if it exceeds the current greedy action value estimate and is deterministic. $Q_{tot}(s, \boldsymbol{\tau}, \mathbf{u}; \theta)$ is trained by minimising the following loss:

$$\sum_{b=1}^{B}(Q_{tot}(s, \boldsymbol{\tau}, \mathbf{u}; \theta) - Q_b^f(s, \boldsymbol{\tau}, \mathbf{u}))^2. \tag{8}$$

Monotonic value factorisation with the original target joint $Q$-values $Q$ induces representational limitation, but it is unclear if the shaped targets do the same. In the following, we give a profound understanding of this. We consider only a fully-observable setting for ease of representation, and our notations do not distinguish the concepts of states and observation-action histories.

**Theorem 2** *Let $Q$ and $Q^f$ be the orignal and the shaped action value based on (7). Then $\forall s \in S$ and $\forall \boldsymbol{u} \in U$ such that $\arg\max_{\boldsymbol{u}} Q(s, \boldsymbol{u})$ is unique, $\arg\max_{\boldsymbol{u}} Q^f(s, \boldsymbol{u}) = \arg\max_{\boldsymbol{u}} Q(s, \boldsymbol{u})$ and $Q^f(s, \boldsymbol{u}) \in \mathcal{Q}^{mvf}$.*

The proof can be found in Appendix A.3. Since all shaped targets $Q^f(s, \mathbf{u})$ at a given state $s$ lie in the restricted monotonic space $\mathcal{Q}^{mvf}$, monotonic value factorisation can perfectly represent these values. In addition, the optimal policy remains unchanged during shaping. Therefore, QMIX can now converge to the optimal for any original target $Q$-value function.

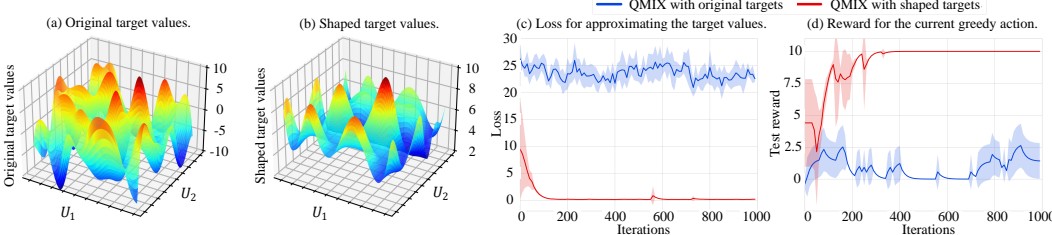

Figure 2: The performance of QMIX with the original and the shaped target values.

In Figure 2, we compare the performance of QMIX with the original and the shaped target joint $Q$-values. Consider a $10 \times 10$ matrix game, where the suboptimal is filled with random numbers generated uniformly between -10 and 9, and the unique optimal value is +10. As shown in Figure 2a-b, the original targets are the rewards from this matrix game, and the shaped targets are obtained based on 7. Figure 2c-d demonstrates that QMIX can perfectly represent all shaped targets and thus converge to the optimal by showing the loss is almost 0 and the test reward is +10 after 400 iterations. By contrast, QMIX with the original targets in Figure 2a keeps fluctuating during training.

## 5 RELATED WORK

In this section, we briefly introduce recent related work on cooperative multi-agent reinforcement learning (MARL) in the paradigm of centralised training with decentralised execution (CTDE). One of the most significant challenges in CTDE is to ensure the correspondence between the individual $Q$-value functions and the joint $Q$-value function $Q_{tot}$, i.e., the Individual-Global Max (IGM) principle (Son et al., 2019). VDN (Sunehag et al., 2018) and QMIX (Rashid et al., 2018) learn the joint $Q$-values and factorise them into individual $Q$-value functions in an additive and a monotonic fashion, respectively. Qatten (Yang et al., 2020b) is a variant of QMIX, which applies a multi-head attention structure to the mixing network. QPD (Yang et al., 2020a) utilises integrated gradients to decompose $Q_{tot}$ along trajectory paths. SMIX($\lambda$) (Yao et al., 2021) changes the one-step $Q$-learning target with a SARSA($\lambda$) target for QMIX. RODE (Wang et al., 2020c) decomposes joint action

spaces into restricted role action spaces to boost learning efficiency and policy generalisation. These methods apply the same monotonic mixing network and thus can only represent the same class as QMIX. However, as many previous studies pointed out, monotonic value factorisation limits the representational capacity of $Q_{tot}$, and fail to learn the optimal policy when the target $Q$-value functions are non-monotonic (Mahajan et al., 2019; Son et al., 2019; Rashid et al., 2020).

Some recent works try to achieve the full representational capacity of $Q_{tot}$ to solve this problem. QPLEX (Wang et al., 2020a) achieves the full representation under the IGM principle theoretically through a duelling mixing network, in which the weights are produced through joint actions. However, it also induces the "vanishing gradient" problem, leading to slow convergence and learning instability. More discussions can be found in Appendix B.2. Deep coordination graph (Böhmer et al., 2020) decompose the joint state-action value into individual utilities and payoff contributions based on the actions of the agents connected by the (hyper-)edges. Tesseract (Mahajan et al., 2021) decomposes the $Q$-tensor across agents and utilises low-rank tensor approximations to model agent interactions relevant to the task, and thus learns a compact approximation of the target $Q$-value function. However, since the dimension of the state-action space increases exponentially as the number of agents grows, it is not easy to achieve the full representational capacity in complex MARL tasks.

Another solution is to learn a biased $Q_{tot}$ by prioritising the optimal joint action. QTRAN (Son et al., 2019) uses two soft regularisations to align the greedy action selections between the joint $Q$-value and the individual values. WQMIX (Rashid et al., 2020) introduces a weighting mechanism similar to Hysteretic $Q$-learning (Matignon et al., 2007), which places more importance on better joint actions. QTRAN can be viewed as a variant of WQMIX, which additionally uses the weight 0 for the joint $Q$-value whose target is smaller than the current estimate and the chosen action is not greedy. QTRAN approximates $\hat{Q}^*$ as the target instead of the original target $y$ and thus can deal with stochasticity. However, due to the 0 weight for overestimated Q-values, QTRAN is empirically hard to scale to more complex tasks (Samvelyan et al., 2019). We also prove that there may not exist an appropriate weight when the targets are non-monotonic and stochastic.

In addition, there have been many developments in policy-based methods under CTDE settings. MAPPO (Yu et al., 2021) applies PPO (Schulman et al., 2017) into MARL and shows strong empirical performance. However, Kuba et al. (2021) points out MAPPO suffers from instability arising from the non-stationarity induced by simultaneously learning and exploring agents. Therefore, they introduce the multi-agent advantage decomposition lemma and the sequential policy update scheme to achieve monotonic improvement on the joint policy, which can converge to one of the Nash Equilibriums. However, the monotonic target in value decomposition methods can be interpreted as a static multivariable optimisation problem, where there are non-optimal solutions that cannot be improved upon by coordinate descent. The sequential policy update scheme cannot guarantee convergence to the optimal in this optimisation problem (Bertsekas, 2019).

**Relationship To Reward Shaping.** Reward shaping is a common technique for improving single-agent learners' performance by integrating expert knowledge into MDP (Gullapalli & Barto, 1992). It also has been adopted to the MARL setting in recent work (Devlin et al., 2011; Xiao et al., 2021). Since it is not always possible to find an expert with complete domain knowledge, it is not easy to set rewards for complex MARL tasks in advance. Therefore, adaptively shaping the reward is a more attractive way. Some reward randomisation methods (Gupta et al., 2021; Tang et al., 2021) are proposed to convert the original MDP to a new MDP with random rewards through universal successor features. However, these methods require a massive number of randomisation to find a desirable reward function because they cannot guarantee *policy invariance*. GVR (Wan et al., 2022) proposes inferior target reshaping and superior experience replay to eliminate the non-optimal self-transition nodes, which is similar to WQMIX and ignores the stochasticity of the transition. Compared with these methods, UTS is more efficient because it can perfectly represent the expected value for stochastic targets and guarantee policy invariance during target shaping.

## 6 EXPERIMENTS

In this section, we conduct empirical experiments to answer the following questions: (1) Is UTS better than the existing methods when the target is seriously non-monotonic? (2) Can UTS perform well when the target is non-monotonic and stochastic? (3) Can UTS perform efficient coordination in challenging multi-agent tasks? We compare UTS with value decomposition baselines, including

VDN (Sunehag et al., 2018), QMIX (Rashid et al., 2018), QTRAN (Son et al., 2019), QPLEX (Wang et al., 2020a) and WQMIX (Rashid et al., 2020) on a matrix game, a multi-agent Markov Decision Process, predator-prey (Son et al., 2019), predator-stag-hare, and StarCraft II Multi-agent Challenge (Samvelyan et al., 2019). We also conduct ablation studies (Appendix C) and compare UTS with GVR (Wan et al., 2022) and WQMIX with different weights (Appendix D), as well as more CTDE-based algorithms (Appendix E). More detailed experimental settings are included in Appendix F.

## 6.1 MATRIX GAMES

In this subsection, we consider a matrix game in Figure 3a, where the local optimal is difficult to jump out due to the considerable miscoordination penalties and stochastic reward. The optimal joint strategy is to perform $(u_1^1, u_2^1)$ simultaneously. We adopt a uniform exploration strategy to approximate the uniform data distribution and eliminate the challenge of exploration and sample complexity. As shown in Figure 3a, UTS and QTRAN can achieve optimal performance, while other algorithms fall into a local optimum. CW-QMIX fluctuates dramatically and cannot converge to any policy, which is consistent with our analysis in Section 3.2.

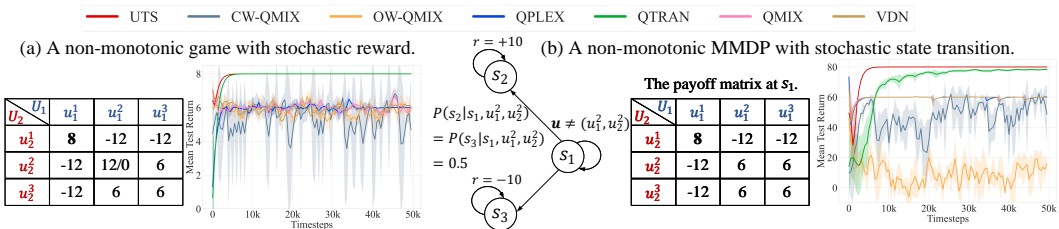

Figure 3: Mean test return on a stochastic matrix game and an MMDP.

We also consider a multi-agent Markov Decision Process (MMDP) with non-monotonic and stochastic targets in Figure 3b. This MMDP involves two agents, three actions and a team reward. Two agents start at state $s_1$ and explore extrinsic rewards for ten environment steps. The state transits from $s_1$ to $s_2$ with probability 50% and $s_3$ with 50% if agents perform $\mathbf{u} = (u_1^2, u_2^2)$ at $s_1$, and remains unchanged otherwise. The team continually obtains a deterministic reward of +10 and -10 at $s_2$ and $s_3$, respectively. In this task, since VDN, QMIX, and QPLEX estimates the stochastic targets well but cannot deal with the non-monotonicity, they converge to the suboptimal with the return of +60. The negative results for CW-QMIX and OW-QMIX show that they converge to the stochastic suboptimal due to the undesirable weights. By contrast, UTS and QTRAN can achieve optimal performance.

## 6.2 PREDATOR-PREY

Predator-prey (PP) is a partially observable environment containing eight predators (agents) and eight prey in a $10 \times 10$ grid world. Each agent observes a $5 \times 5$ sub-grid around it and can perform five actions, i.e., up, down, left, right, and catch. The team receives a miscoordination penalty when two agents surround a prey and only one tries to catch it. By contrast, they earn a bonus of 10 if they catch the prey simultaneously. After a successful catch, the catching predators and prey will be removed from the grid. Figure 4a-b shows that VDN, QMIX, and QPLEX fail to return positive returns with a significant penalty because the penalty makes these methods underestimate the per-agent action value of the real optimal. CW-QMIX and OW-QMIX can achieve positive test returns when the penalty is -2 because they place more weight on better joint actions. QTRAN can also solve this task through its regularisation terms. However, CW-QMIX, OW-QMIX, and QTRAN fail to solve the task when the penalty is -10, suggesting difficulties in choosing the pre-defined weight and the network architecture. By contrast, UTS solves tasks quickly and steadily in all penalty settings.

We extend predator-prey to a stochastic version named predator-stag-hare (PSH), which contains eight predators, four stags and four hares. The agents receive the same miscoordination penalty in predator-prey when they try to catch a stag. By contrast, a catch of hare results in a non-deterministic reward, i.e., the probabilities of receiving SR$\in \mathbb{R}^+$ and -SR are both 50%. As shown in Figure 4c-d, VDN and QMIX can quickly recover the optimal policy when the penalty is 0 because they can

accurately evaluate the stochastic reward. When the penalty is -2, VDN, QMIX, QPLEX, and WQMIX fail to solve this task. QTRAN also struggles with the suboptimal in this setting. By contrast, UTS takes advantage of the uncertainty estimation module to avoid overestimating the stochastic action values, and uses the shaping function to eliminate the adverse impact of the miscoordination penalty. Therefore, it learns quickly and reliably in these tasks. The detailed comparison between UTS and WQMIX with different weights can be found in Appendix D.

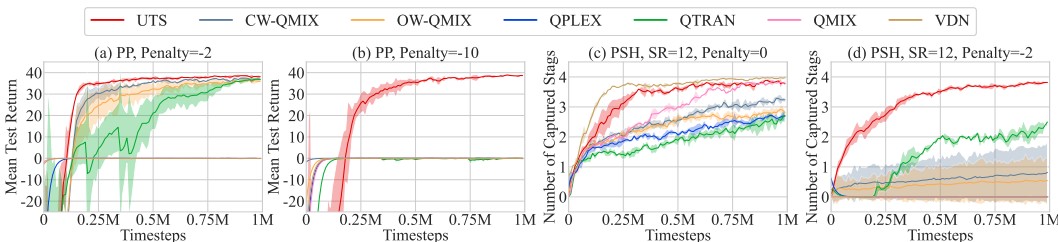

Figure 4: Mean test return on predator-prey and predator-stag-hare.

### 6.3 STARCRAFT MULTI-AGENT CHALLENGE

We also compare UTS with baselines on eight hard and super hard maps in the decentralised micromanagement tasks in StarCraft Multi-Agent Challenge (SMAC) (Samvelyan et al., 2019). Figure 5 shows the improved performance of UTS, which indicates that the target shaping function can be applied to more general benchmarks.

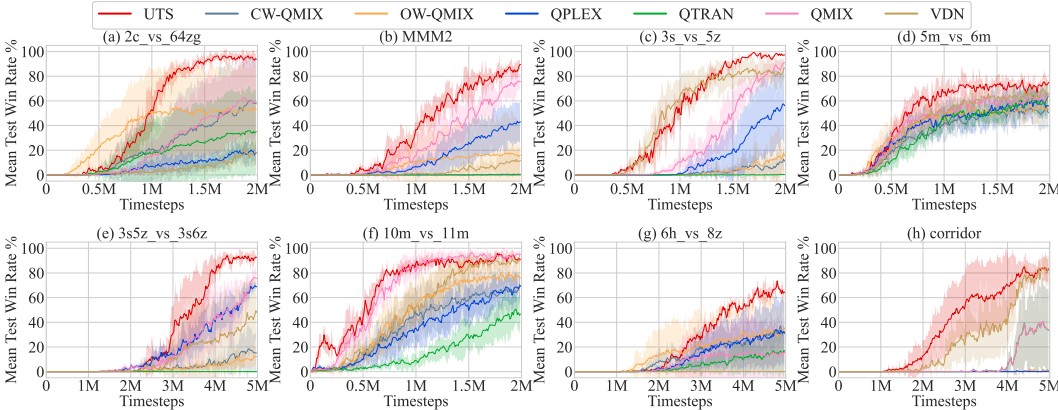

Figure 5: Mean test win rate on SMAC tasks.

## 7 CONCLUSION AND FUTURE WORK

This paper presents uncertainty-based target shaping (UTS), which was inspired by analysing the limitation of Weighted QMIX. We prove that there may not exist an appropriate weighting function for Weighted QMIX to recover the optimal policy when the targets are stochastic and non-monotonic. To solve this problem, UTS first identifies whether a target value is stochastic by predicting the standard deviation of the reward and the prediction error between the predicted and the target embedding of the following state. Then we propose a shaping function to project original target values into the monotonic space that QMIX can represent by replacing the deterministic suboptimal target with the minimal best per-agent values. Empirical results demonstrate the improved ability of UTS to deal with the task with non-monotonic and stochastic targets, as well as more complicated tasks. The assumption of "unique optimal policy" is required in our proofs. In the environment with multiple optimal, the more efficient exploration method could be considered, as opposed to the simplistic $\epsilon$-greedy exploration scheme we used in this paper.

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

# A    OMITTED PROOFS

## A.1    WVDN AND WEIGHTED LINEAR REGRESSION PROBLEM

WVDN operator is defined as

$$\Pi_{\text{WVDN}}Q := \underset{Q_{tot} \in \mathcal{Q}^{lvf}}{\arg\min} \sum_{\mathbf{u} \in \mathbf{U}} w^{ow}(s, \mathbf{u})(y(s, \mathbf{u}) - Q_{tot}(s, \mathbf{u}))^2, \tag{9}$$

where $\mathcal{Q}^{lvf} := \{Q_{tot}|Q_{tot}(s, \mathbf{u}) = \sum_{a=1}^{n} Q_a(s, u_a), Q_a(s, u) \in \mathbb{R}\}$, $w^{ow}(s, \mathbf{u})$ is the optimistic weighting:

$$w^{ow}(s, \mathbf{u}) = \begin{cases} 1 & y(s, \mathbf{u}) > Q_{tot}(s, \mathbf{u}) \\ \alpha & \text{otherwise} \end{cases}. \tag{10}$$

The optimising process of WVDN can be constructed as a weighted linear regression problem:

$$\min_{\mathbf{x}} \|\sqrt{\mathbf{w}^\top} \cdot \sqrt{\mathbf{p}^\top} \cdot (\mathbf{A}\mathbf{x} - \mathbf{b})\|, \tag{11}$$

where $\mathbf{A} \in \mathbb{R}^{m^n \times mn}$, $\mathbf{x} \in \mathbb{R}^{mn}$, $\mathbf{w}, \mathbf{p}, \mathbf{b} \in \mathbb{R}^{m^n}$, $m, n \in Z^+$. In particular, $m$ dentoes the action space size for each agent, $n$ denotes the number of agents.

Denote $\mathbf{A}^w = \sqrt{\mathbf{w}^\top} \cdot \sqrt{\mathbf{p}^\top} \cdot \mathbf{A}$, $\mathbf{b}^w = \sqrt{\mathbf{w}^\top} \cdot \sqrt{\mathbf{p}^\top} \cdot \mathbf{b}$. The optimal solution is:

$$\mathbf{x}^* = \mathbf{A}^{w,\dagger}\mathbf{b}^w, \tag{12}$$

where $\mathbf{A}^{w,\dagger} = VS^\dagger U^\top$ is the pseudo-inverse of $\mathbf{A}^w$ according to the Singular Value Decomposition. $U \in \mathbb{R}^{m^n \times m^n}$ and $V \in \mathbb{R}^{mn \times mn}$ are real orthogonal matrices, $S^\dagger \in \mathbb{R}^{m^n \times mn}$ is a diagonal matric with non-negative real numbers on the diagonal.

The proof is trivial.

Consider the full exploration scheme (i.e., $\epsilon$-greedy exploration with $\epsilon = 1$), and the non-monotonic game in Figure 6a, where $a = 8, b = -12, c = 6$.

For VDN (a special case of WVDN where $\alpha = 1$), we have

$$\mathbf{A} = \begin{pmatrix} 1 & 0 & 0 & 1 & 0 & 0 \\ 1 & 0 & 0 & 0 & 1 & 0 \\ 1 & 0 & 0 & 0 & 0 & 1 \\ 0 & 1 & 0 & 1 & 0 & 0 \\ 0 & 1 & 0 & 0 & 1 & 0 \\ 0 & 1 & 0 & 0 & 0 & 1 \\ 0 & 0 & 1 & 1 & 0 & 0 \\ 0 & 0 & 1 & 0 & 1 & 0 \\ 0 & 0 & 1 & 0 & 0 & 1 \end{pmatrix}, \mathbf{x} = \begin{pmatrix} Q_1(s, u_1^1) \\ Q_1(s, u_1^2) \\ Q_1(s, u_1^3) \\ Q_2(s, u_2^1) \\ Q_2(s, u_2^2) \\ Q_2(s, u_2^3) \end{pmatrix}, \mathbf{b} = \begin{pmatrix} 8 \\ -12 \\ -12 \\ -12 \\ 6 \\ 6 \\ -12 \\ 6 \\ 6 \end{pmatrix}, \mathbf{p} = \mathbf{w} = \begin{pmatrix} 1 \\ 1 \\ 1 \\ 1 \\ 1 \\ 1 \\ 1 \\ 1 \\ 1 \end{pmatrix}. \quad (13)$$

According to (12), the optimal solution is

$$\mathbf{x}^* = \begin{pmatrix} -4.44 & 0.89 & 0.89 & -4.44 & 0.89 & 0.89 \end{pmatrix}^\top. \quad (14)$$

Clearly, $Q_i(u_i^1) < \max\{Q_i(u_i^2), Q_1(u_i^3)\}, \forall i \in \{1, 2\}$, which indicates that VDN fails to converge to the optimal.

## A.2 OMITTED PROOFS IN SECTION 3

WVDN operator is defined in Section A.1.

Denote the joint action $\mathbf{u} = [u_1, ..., u_a, ..., u_n]$, where $n$ is the number of agents. Denote $u_a^i$ as the $i$-th action of $u_a$.

Consider matrix games in Figure 6. Suppose the training dataset is fixed and the data has a $\epsilon$-greedy distribution, where the greedy action is $(u_1^2, u_2^2)$ and $\epsilon \in (0, 1]$.

Based on the $\epsilon$-greedy distribution, the probabilities of each joint action $p(u_1^i, u_2^j)$ is:

$$Pr(u_1^i, u_2^j) = \begin{cases} \dfrac{\epsilon^2}{9}, & (i, j) \in \{(1,1), (1,3), (3,1), (3,3)\}, \\ \dfrac{(3 - 2\epsilon)^2}{9}, & (i, j) = (2, 2), \\ \dfrac{(3 - 2\epsilon)\epsilon}{9}, & \text{otherwise.} \end{cases} \quad (15)$$

| $U_2$\$U_1$ | $u_1^1$ | $u_1^2$ | $u_1^3$ |
|---|---|---|---|
| $u_2^1$ | **a** | b | b |
| $u_2^2$ | b | c | c |
| $u_2^3$ | b | c | c |

(a) A non-monotonic matrix game.

| $U_2$\$U_1$ | $u_1^1$ | $u_1^2$ | $u_1^3$ |
|---|---|---|---|
| $u_2^1$ | **a** | b | b |
| $u_2^2$ | b | c/d | e |
| $u_2^3$ | b | e | e |

(b) A stochastic matrix game.

| $U_2$\$U_1$ | $u_1^1$ | $u_1^2$ | $u_1^3$ |
|---|---|---|---|
| $u_2^1$ | 8 | 6 | 6 |
| $u_2^2$ | 6 | 12/0 | 6 |
| $u_2^3$ | 6 | 6 | 6 |

(c) Payoff of a stochastic matrix game.

| $U_2$\$U_1$ | $u_1^1$ | $u_1^2$ | $u_1^3$ |
|---|---|---|---|
| $u_2^1$ | **8** | -12 | -12 |
| $u_2^2$ | -12 | 12/0 | 6 |
| $u_2^3$ | -12 | 6 | 6 |

(d) Payoff of a stochastic and non-monotonic game.

Figure 6: Payoffs of matrix games. (a) is a non-monotonic matrix game, where $a > c > b$; (b) is a stochastic matrix game, where $(u_1^2, u_2^2)$ receives $c$ with probability $p$ and $d$ with $(1 - p)$, $c > a > \max\{b, d, e\}$ and $a > pc + (1 - p)d$; (c-d) are two numerical examples for stochastic matrix games, where the reward in (c) is monotonic and in (d) is non-monotonic.

To recover the optimal policy, the weight $\alpha$ for WVDN is bounded by $\epsilon$ and the reward space. We start by analysing the matrix game in Figure 6a. Since the matrix game only involves one state, we omit $s$ for the input of value functions in the following proofs.

**Proposition 1** *For WVDN, the weight $\alpha$ should be smaller than $\frac{3\epsilon(a-c)+\epsilon^2(c-b)}{(\epsilon-3)^2(c-b)}$ to recover the optimal joint policy on the non-monotonic matrix game in Figure 6a.*

*Proof.* First compute the gradient for $Q(u_1^1), Q(u_1^2), Q(u_1^3)$:

$$\nabla Q(u_1^1) = \frac{\epsilon}{9}[(\epsilon + (3-\epsilon)\alpha)Q(u_1^1) + \epsilon Q(u_2^1) + (3-2\epsilon)\alpha Q(u_2^2) + \epsilon\alpha Q(u_2^3)$$
$$- \epsilon a - \alpha(3-\epsilon)b] \tag{16}$$

$$\nabla Q(u_1^2) = \frac{(3-2\epsilon)\alpha}{9}[3Q(u_1^2) + \epsilon(Q(u_2^1) + Q(u_2^3)) + (3-2\epsilon)Q(u_2^2) - \epsilon b - (3-\epsilon)c] \tag{17}$$

$$\nabla Q(u_1^3) = \frac{\epsilon\alpha}{9}[3Q(u_1^3) + \epsilon(Q(u_2^1) + Q(u_2^3)) + (3-2\epsilon)Q(u_2^2) - \epsilon b - (3-\epsilon)c] \tag{18}$$

Let $\nabla Q(u_1^1) = \nabla Q(u_1^2) = \nabla Q(u_1^3) = 0$:

$$Q(u_1^1) = \frac{\epsilon a + \alpha(3-\epsilon)b - \epsilon Q(u_2^1) - (3-2\epsilon)\alpha Q(u_2^2) - \epsilon\alpha Q(u_2^3)}{\epsilon + 3\alpha - \epsilon\alpha} \tag{19}$$

$$Q(u_1^2) = Q(u_1^3) = \frac{\epsilon b + (3-\epsilon)c - \epsilon(Q(u_2^1) + Q(u_2^3)) - (3-2\epsilon)Q(u_2^2)}{3} \tag{20}$$

To recover the optimal joint policy, the following conditions should be satisfied:

$$Q(u_1^1) - \max\{Q(u_1^2), Q(u_1^3)\} > 0, Q(u_2^1) - \max\{Q(u_2^2), Q(u_2^3)\} > 0 \tag{21}$$

Denote $t = \epsilon + 3\alpha - \epsilon\alpha$. Then

$$Q(u_1^1) - \max\{Q(u_1^2), Q(u_1^3)\}$$
$$= \frac{\epsilon a + \alpha(3-\epsilon)b - \epsilon Q(u_2^1) - (3-2\epsilon)\alpha Q(u_2^2) - \epsilon\alpha Q(u_2^3)}{t}$$
$$- \frac{\epsilon b + (3-\epsilon)c - \epsilon(Q(u_2^1) + Q(u_2^3)) - (3-2\epsilon)Q(u_2^2)}{3}$$
$$< \underbrace{(\frac{t + \epsilon\alpha - \epsilon - 3\alpha}{t})}_{= 0}Q(u_2^1) + \frac{\epsilon}{t}a + \frac{(9-3\epsilon)\alpha - \epsilon t}{3t}b + \frac{\epsilon - 3}{3}c. \tag{22}$$

Let $Q(u_1^1) - \max\{Q(u_1^2), Q(u_1^3)\} > 0$ and $Q(u_2^1) - \max\{Q(u_2^2), Q(u_2^3)\} > 0$.

Then $\alpha < \frac{3\epsilon(a-c) + \epsilon^2(c-b)}{(\epsilon-3)^2(c-b)}$. $\square$

**Proposition 2** *For WVDN, the weight $\alpha$ should be large enough to recover the optimal joint policy for the stochastic matrix game in Figure 6b.*

**Theorem 1** *Let $\Pi_{\text{WVDN}}Q := \arg\min_{Q_{tot} \in \mathcal{Q}^{lvf}} \sum_{\boldsymbol{u} \in \mathbf{U}} w^{ow}(s, \boldsymbol{u})(Q(s, \boldsymbol{u}) - Q_{tot}(s, \boldsymbol{u}))^2$ and $w^{ow}(s, \boldsymbol{u})$ is the optimistic weighting from (4). Then $\exists Q$ such that $\arg\max \Pi_{\text{WVDN}}Q \neq \arg\max Q$ for any $\alpha \in (0, 1]$.*

*Proof.* Since the proof of Proposition 2 and Theorem 1 contains a significant overlap, we will merge them both into a single proof.

First compute the gradient for $Q(u_1^1), Q(u_1^2), Q(u_1^3)$:

$$\nabla Q(u_1^1) = \frac{\epsilon}{9}[(\epsilon + 3\alpha - \epsilon\alpha)Q(u_1^1) + \epsilon Q(u_2^1) + (3-2\epsilon)\alpha Q(u_2^2) + \epsilon\alpha Q(u_2^3) -$$
$$\epsilon a - \alpha(3-\epsilon)b] \tag{23}$$

$$\nabla Q(u_1^2) = \frac{3-2\epsilon}{9}[((3-2\epsilon)(1-\alpha)p + 3\alpha)Q(u_1^2) + \epsilon\alpha(Q(u_2^1) + Q(u_2^3))$$
$$+ (3-2\epsilon)(p + \alpha - p\alpha)Q(u_2^2) - \epsilon\alpha b - (3-2\epsilon)(pc + \alpha d - \alpha pd) - \epsilon\alpha e] \tag{24}$$

$$\nabla Q(u_1^3) = \frac{\epsilon\alpha}{9}[3Q(u_1^3) + \epsilon(Q(u_2^1) + Q(u_2^3)) + (3-2\epsilon)Q(u_2^2) - \epsilon b - (3-\epsilon)e] \tag{25}$$

To recover the optimal joint policy, the following requirements should be satisfied:

$$Q(u_1^1) - \max\{Q(u_1^2), Q(u_1^3)\} > 0, Q(u_2^1) - \max\{Q(u_2^2), Q(u_2^3)\} > 0$$

Let $\nabla Q(u_1^1) = \nabla Q(u_1^2) = \nabla Q(u_1^3) = 0$. Denote $t = \epsilon + 3\alpha - \epsilon\alpha$, $m = (3 - 2\epsilon)(1 - \alpha)p + 3\alpha$. Then

$$Q(u_1^1) - Q(u_1^2) < \frac{\epsilon}{t}a + \frac{(3-\epsilon)\alpha}{t}b - \frac{\epsilon\alpha}{m}(b+e) - \frac{(3-2\epsilon)p}{m}c - \frac{(3-2\epsilon)(1-p)\alpha}{m}d, \quad (26)$$

$$Q(u_1^1) - Q(u_1^3) < \frac{\epsilon}{t}a + \frac{(3-\epsilon)\alpha}{t}b - \frac{\epsilon}{3}b - \frac{(3-\epsilon)p}{3}e. \quad (27)$$

For the matrix game in Figure 6b, consider the following situations:

(1) The target is monotonic but stochastic. For example, in Figure 6c, $a = 8$, $b = e = 6$, $c = 12$, $d = 0$, and $p = 0.5$. To recover the optimal joint action, the weight $\alpha$ should be greater than $(0.434, 0.667, 0.827, 0.932)$ when $\epsilon = (1, 0.75, 0.5, 0.25)$, respectively.

(2) The target is non-monotonic and stochastic. For example, in Figure 6d, $a = 8$, $b = -12$, $c = 12$, $d = 0$, $e = 6$, and $p = 0.5$. When $\epsilon = (1, 0.75, 0.5, 0.25)$, there does not exist the weight $\alpha \in (0, 1]$ making $Q(u_1^1) - \max\{Q(u_1^2), Q(u_1^3)\} > 0$, i.e., WVDN cannot converge to the optimal.

Thus, $\exists Q$ such that $\arg\max \Pi_{\text{WVDN}} Q \neq \arg\max Q$ for any $\alpha \in (0, 1]$. $\square$

More experimental results can be found in Appendix B.1.

### A.3 OMITTED PROOFS OF RESHAPING FUNCTION

**Theorem 2** *Let $Q$ and $Q^f$ be the orignal and the shaped action value based on* (7). *Then $\forall s \in S$ and $\forall \boldsymbol{u} \in U$ such that $\arg\max_{\boldsymbol{u}} Q(s, \boldsymbol{u})$ is unique, $\arg\max_{\boldsymbol{u}} Q^f(s, \boldsymbol{u}) = \arg\max_{\boldsymbol{u}} Q(s, \boldsymbol{u})$ and $Q^f(s, \boldsymbol{u}) \in \mathcal{Q}^{mvf}$.*

*Proof.* Considering only a fully-observable setting for ease of representation. Thus our notations do not distinguish the concepts of states and observation-action histories. In addition, when more than one optimal policy exists, most $Q$-learning algorithms fail to converge to a stable point (Simchowitz & Jamieson, 2019; Wang et al., 2020a; Rashid et al., 2020). Thus, consider a $s \in S$, we assume $\mathbf{u}_Q^* = \arg\max_{\mathbf{u}} Q(s, \mathbf{u})$ is unique.

Assume $\max_{\mathbf{u}_{-a}} Q(s, u_a, \mathbf{u}_{-a})$ for all $u_a$ are perfectly represented.

Then $Q(s, \mathbf{u}) \leqslant \min_{a \in \mathbb{A}}\{\max_{\mathbf{u}_{-a}} Q(s, u_a, \mathbf{u}_{-a})\}$.

By the shaping function from (7):

$$\begin{cases} Q^f(s, \mathbf{u}_Q^*) = Q(s, \mathbf{u}_Q^*) \\ Q^f(s, \mathbf{u}) = \min\{\max_{\mathbf{u}_{-i}} Q(s, u_i, \mathbf{u}_{-i})\}, \forall u_i \in \mathbf{u} \neq \mathbf{u}_Q^* \text{ and } u_i \notin \mathbf{u}_Q^* \end{cases} \quad (28)$$

Under the assumption of the unique optimal action, we have $Q^f(s, \mathbf{u}_Q^*) > Q^f(s, \mathbf{u})$ for all $\mathbf{u} \neq \mathbf{u}_Q^*$. Therefore, $\arg\max_{\mathbf{u}} Q^f(s, \mathbf{u}) = \arg\max_{\mathbf{u}} Q(s, \mathbf{u})$.

Suppose $\exists f_s(s, Q_1(s, u_1), ..., Q_n(s, u_n)) = Q^f(s, \mathbf{u}), \forall \mathbf{u} = (u_1, ..., u_n) \in U$. Then

$$\begin{cases} f_s(s, Q_a(u_a), \boldsymbol{Q}_{-a}(\mathbf{u}_{-a})) = \min\{\max_{\mathbf{u}_{-a}} Q(s, u_a, \mathbf{u}_{-a}), \max_{\mathbf{u}_{-i}} Q(s, u_i, \mathbf{u}_{-i})\}, \forall i \in -a \\ f_s(s, Q_a(u_a'), \boldsymbol{Q}_{-a}(\mathbf{u}_{-a})) = \min\{\max_{\mathbf{u}_{-a}} Q(s, u_a', \mathbf{u}_{-a}), \max_{\mathbf{u}_{-i}} Q(s, u_i, \mathbf{u}_{-i})\}, \forall i \in -a \end{cases}, \quad (29)$$

where $\boldsymbol{Q}_{-a}(\mathbf{u}_{-a}) = \{Q_i(s, u_i)\}_{i \in -a}$.

Let $Q_a(s, \mathbf{u}_a) \geqslant Q_a(s, \mathbf{u}_a')$ if $\max_{\mathbf{u}_{-a}} Q(s, u_a, \mathbf{u}_{-a}) \geqslant \max_{\mathbf{u}_{-a}} Q(s, u_a', \mathbf{u}_{-a}), \forall a \in A$.

Then, according to (29), $f_s(s, Q_a(u_a), \boldsymbol{Q}_{-a}(\mathbf{u}_{-a})) \geqslant f_s(s, Q_a(u_a'), \boldsymbol{Q}_{-a}(\mathbf{u}_{-a}))$.

Thus we can find $f_s(s, Q_1(s, u_1), ..., Q_n(s, u_n))$ to represent $Q^f(s, \mathbf{u})$, $\forall \mathbf{u} = (u_1, ..., u_n) \in U$, where $\frac{\partial f_s}{\partial Q_a} \geqslant 0$. And so $Q^f(s, \mathbf{u}) \in \mathcal{Q}^{mvf}, \forall s \in S, \forall \mathbf{u} \in U$. $\square$

Now we talk about the case of multiple optimal. When more than one optimal policy exists in the Markov Decision Process, many value-based algorithms fail to converge on a stable point (Simchowitz & Jamieson, 2019). However, multiple optimal may be very common in a multi-agent setting. For

example, when two agents arrive at a crossroads, the optimal solution is that one of them chooses to move and another chooses to make way. In this situation, the team can solve this simple task no matter which agent chooses to move.

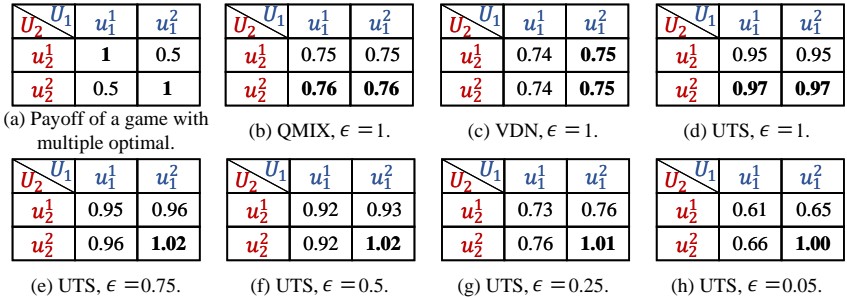

| $U_2$ \ $U_1$ | $u_1^1$ | $u_1^2$ |
|---|---|---|
| $u_2^1$ | **1** | 0.5 |
| $u_2^2$ | 0.5 | **1** |

(a) Payoff of a game with multiple optimal.

| $U_2$ \ $U_1$ | $u_1^1$ | $u_1^2$ |
|---|---|---|
| $u_2^1$ | 0.75 | 0.75 |
| $u_2^2$ | **0.76** | **0.76** |

(b) QMIX, $\epsilon = 1$.

| $U_2$ \ $U_1$ | $u_1^1$ | $u_1^2$ |
|---|---|---|
| $u_2^1$ | 0.74 | **0.75** |
| $u_2^2$ | 0.74 | **0.75** |

(c) VDN, $\epsilon = 1$.

| $U_2$ \ $U_1$ | $u_1^1$ | $u_1^2$ |
|---|---|---|
| $u_2^1$ | 0.95 | 0.95 |
| $u_2^2$ | **0.97** | **0.97** |

(d) UTS, $\epsilon = 1$.

| $U_2$ \ $U_1$ | $u_1^1$ | $u_1^2$ |
|---|---|---|
| $u_2^1$ | 0.95 | 0.96 |
| $u_2^2$ | 0.96 | **1.02** |

(e) UTS, $\epsilon = 0.75$.

| $U_2$ \ $U_1$ | $u_1^1$ | $u_1^2$ |
|---|---|---|
| $u_2^1$ | 0.92 | 0.93 |
| $u_2^2$ | 0.92 | **1.02** |

(f) UTS, $\epsilon = 0.5$.

| $U_2$ \ $U_1$ | $u_1^1$ | $u_1^2$ |
|---|---|---|
| $u_2^1$ | 0.73 | 0.76 |
| $u_2^2$ | 0.76 | **1.01** |

(g) UTS, $\epsilon = 0.25$.

| $U_2$ \ $U_1$ | $u_1^1$ | $u_1^2$ |
|---|---|---|
| $u_2^1$ | 0.61 | 0.65 |
| $u_2^2$ | 0.66 | **1.00** |

(h) UTS, $\epsilon = 0.05$.

Figure 7: (a) The payoff matrix for a game with multiple optimal. (b-h) The estimated joint $Q$-values $Q_{tot}$ returned from QMIX, VDN, and UTS with different exploration rate. Boldface means the optimal joint action from the payoff matrix, or the greedy joint action from the $Q_{tot}$.

We take a simple one-step matrix game with multiple optimal as an example. The payoff matrix is shown in Figure 7a. We investigate the performance of VDN, QMIX, QPLEX, and UTS on this simple task. Since this matrix game is non-monotonic, the target values cannot be perfectly represented in $\mathcal{Q}^{lvf}$ and $\mathcal{Q}^{mvf}$. When the full exploration scheme (i.e., $\epsilon$-greedy with $\epsilon = 1$) is applied, none of these methods can converge to the optimal. Although QPLEX achieves the full representational capacity (i.e., the joint $Q$-values $Q_{tot}$ is the same as the payoff), it still cannot converge because all per-agent Q-values are optimal (because of the Individual-Global-Max). However, in practice, we can solve this problem by $\epsilon$-greedy with decayed $\epsilon$. In particular, we show the $Q_{tot}$ returned from UTS to illustrate the effect of the $\epsilon$. As shown in Figure 7e-h, we can see the gap between the suboptimal and the optimal increases with the decrease of the $\epsilon$, which improves the learning stability. However, UTS inherits the characteristic of QMIX and thus is not a contraction, i.e., it would return two distinct $Q_{tot}$ and converge to any one of the optimal.

# B   RESULTS ON MATRIX GAMES

## B.1   WQMIX WITH DIFFERENT WEIGHTS

In Section 3.1, we demonstrate the complete data of minimal $1/\alpha$ for WVDN and WQMIX to recover the optimal policy, where $\alpha$ denotes the weight on the suboptimal. The payoff matrix is shown in 6a. The dataset is collected by the $\epsilon$-greedy exploration, where the greedy action is set to $\mathbf{u}^s = (u_1^2, u_2^2)$.

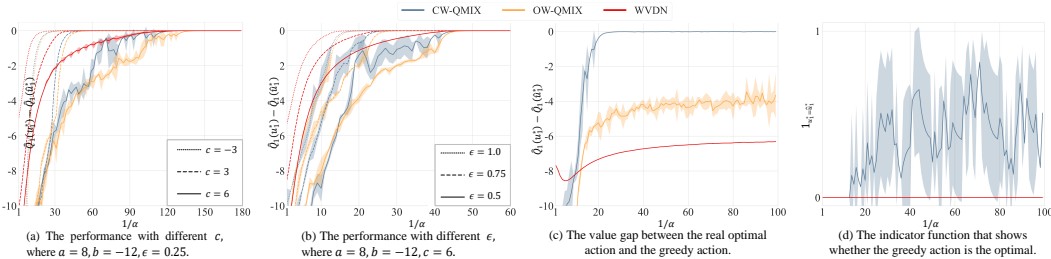

(a) The performance with different $c$, where $a = 8, b = -12, \epsilon = 0.25$.

(b) The performance with different $\epsilon$, where $a = 8, b = -12, c = 6$.

(c) The value gap between the real optimal action and the greedy action.

(d) The indicator function that shows whether the greedy action is the optimal.

Figure 8: (a)-(b) shows the performance of WVDN and WQMIX on the non-monotonic game in Figure 6a, and (c)-(d) shows the performance on the non-monotonic and stochastic game in Figure 9a.

We select $a = 8$, $b = -12$, and (1) three reward assignments, $c \in \{-3, 3, 6\}$, $\epsilon = 0.25$, as well as (2) three exploration rate assignments, $\epsilon \in \{0.5, 0.75, 1\}$, $c = 6$, and run WVDN and WQMIX with five random seeds on a fixed dataset. We show the gap between the approximated real optimal action value $\hat{Q}_1(u_1^*)$ and greedy action value $\hat{Q}_1(\hat{u}_1^*)$ for agent 1, which should be zero if the algorithm converges to the optimal. Figure 8a-b shows that the gap reduces with the increase of $1/\alpha$, indicating

that the weight $\alpha$ should be small enough to deal with non-monotonicity. In addition, we can see that $\alpha$ should be small enough to converge to the optimal when the suboptimal value is large or the exploration rate is low, which is consistent with our analysis.

In Section 3.2 and Section A.2, we prove that the contradiction of the choice of $\alpha$ may result in converging on the suboptimal for WQMIX. We also conduct experiments to verify our analysis. We consider the matrix game in Figure 9a, and train WVDN and WQMIX on a fixed dataset. The dataset is collected by the $\epsilon$-greedy exploration, where the greedy action is set to $\mathbf{u}^s = (u_1^2, u_2^2)$ and $\epsilon = 0.5$. We show the gap between the approximated real optimal action value $\hat{Q}_1(u_1^*)$ and greedy action value $\hat{Q}_1(\hat{u}_1^*)$ for agent 1, which should be zero if the algorithm converges to the optimal. We also show whether the greedy action is optimal by an indicator function $\mathbf{1}_{\hat{Q}_1(\mathbf{u}_1^*)=\hat{Q}_1(\hat{u}_1^*)}$.

As shown in Figure 8c-d, WQMIX and WVDN cannot converge to the optimal with different $\alpha$ when the payoff is non-monotonic and stochastic. Since CW-QMIX does not place $\alpha$ on the current greedy joint action, it can get rid of the stochastic suboptimal $\mathbf{u}^s$ when $\mathbf{u}^s$ is the current greedy action, i.e., $\hat{\mathbf{u}}^* = \mathbf{u}^s$. However, with the decrease of $Q_{tot}(\mathbf{u}^s)$, CW-QMIX falls into this suboptimal again once $\hat{\mathbf{u}}^* \neq \mathbf{u}^s$. Therefore, we can see that the gap between $\hat{Q}_1(u_1^*)$ and $\hat{Q}_1(\hat{u}_1^*)$ is tiny. However, as shown in Figure 8d, CW-QMIX cannot converge to the optimal, which is consistent with our analysis.

### B.2 THE APPROXIMATED JOINT ACTION-VALUES

We also show the joint $Q$-value returned from our baselines on a matrix game with non-monotonic and stochastic rewards in Figure 9a. We adopt a uniform exploration strategy to approximate the uniform data distribution (i.e., $\epsilon$-greedy exploration with $\epsilon = 1$). As shown in Figure 9, UTS and QTRAN can recover the optimal policy, while other methods fail to solve this simple task. Since we have discussed QMIX and WQMIX in previous sections, here we analyse QPLEX and GVR.

| $U_2 \backslash U_1$ | $u_1^1$ | $u_1^2$ | $u_1^3$ |
|---|---|---|---|
| $u_2^1$ | **8** | -12 | -12 |
| $u_2^2$ | -12 | 20 / -20 | 6 |
| $u_2^3$ | -12 | 6 | 6 |

(a) A game with non-monotonic and stochastic rewards.

| $U_2 \backslash U_1$ | $u_1^1$ | $u_1^2$ | $u_1^3$ |
|---|---|---|---|
| $u_2^1$ | **8** | 6 | 6 |
| $u_2^2$ | 6 | 6 | 6 |
| $u_2^3$ | 6 | 6 | 6 |

(b) UTS.

| $U_2 \backslash U_1$ | $u_1^1$ | $u_1^2$ | $u_1^3$ |
|---|---|---|---|
| $u_2^1$ | 1.68 | 1.68 | 1.68 |
| $u_2^2$ | 1.68 | **16.8** | 6.33 |
| $u_2^3$ | 1.68 | 6.33 | 5.93 |

(c) OW-QMIX ($\alpha$=0.1).

| $U_2 \backslash U_1$ | $u_1^1$ | $u_1^2$ | $u_1^3$ |
|---|---|---|---|
| $u_2^1$ | 2.72 | 2.72 | 2.72 |
| $u_2^2$ | 2.72 | **5.97** | **5.97** |
| $u_2^3$ | 2.72 | **5.97** | **5.97** |

(d) CW-QMIX ($\alpha$=0.1).

| $U_2 \backslash U_1$ | $u_1^1$ | $u_1^2$ | $u_1^3$ |
|---|---|---|---|
| $u_2^1$ | -8.29 | -8.29 | -8.29 |
| $u_2^2$ | -8.29 | 1.49 | 4.64 |
| $u_2^3$ | -8.29 | 5.00 | **8.15** |

(e) QMIX ($\alpha$=1).

| $U_2 \backslash U_1$ | $u_1^1$ | $u_1^2$ | $u_1^3$ |
|---|---|---|---|
| $u_2^1$ | 5.31 | 5.33 | 5.33 |
| $u_2^2$ | 5.35 | **6.03** | **6.03** |
| $u_2^3$ | 5.35 | **6.03** | **6.03** |

(f) GVR.

| $U_2 \backslash U_1$ | $u_1^1$ | $u_1^2$ | $u_1^3$ |
|---|---|---|---|
| $u_2^1$ | **8.09** | -12.32 | -12.24 |
| $u_2^2$ | -12.32 | -2.27 | 5.76 |
| $u_2^3$ | -12.24 | 5.76 | 5.62 |

(g) QTRAN.

| $U_2 \backslash U_1$ | $u_1^1$ | $u_1^2$ | $u_1^3$ |
|---|---|---|---|
| $u_2^1$ | 4.37 | -8.06 | -8.45 |
| $u_2^2$ | -6.93 | 1.61 | **4.37** |
| $u_2^3$ | -8.09 | **4.37** | **4.37** |

(h) QPLEX.

Figure 9: The results on a matrix game with non-monotonic and stochastic rewards. Boldface means the optimal joint action from the payoff matrix, or the greedy joint action from the $Q_{tot}$.

In QPLEX, the joint $Q$-value is:

$$Q_{tot}(\boldsymbol{\tau}, \mathbf{u}) = V_{tot}(\boldsymbol{\tau}) + A_{tot}(\boldsymbol{\tau}, \mathbf{u}) = \sum_{i=1}^{n} V_i(\boldsymbol{\tau}) + \sum_{i=1}^{n} \lambda_i(\boldsymbol{\tau}, \mathbf{u}) A_i(\boldsymbol{\tau}, u_i)$$

$$= \sum_{i=1}^{n}[w_i(\boldsymbol{\tau}) V_i(\tau_i) + b_i(\boldsymbol{\tau})] + \sum_{i=1}^{n} \lambda_i(\boldsymbol{\tau}, \mathbf{u})[Q_i(\boldsymbol{\tau}, u_i) - V_i(\boldsymbol{\tau})] \quad (30)$$

where $V_i(\tau_i) = \max_{u_i} Q_i(\tau_i, u_i), w_i(\boldsymbol{\tau}) \geq 0, \lambda_i(\boldsymbol{\tau}, \mathbf{u}) > 0$.

Let $\mathbf{u}^* = [u_1^*, ..., u_i^*, ..., u_n^*]$, $\mathbf{u}^s = [u_1^s, ..., u_i^s, ..., u_n^s]$ denote the optimal and one of the suboptimal joint action, where $n$ is the number of agents. If QPLEX is initialised to this joint action, i.e., $V_{tot}(\boldsymbol{\tau}) = \sum_{i=1}^{n} Q_i(\tau_i, u_i^s)$, it is difficult to jump out from this suboptimal for QPLEX. Consider $Q(s, \mathbf{u}^s)$ is the real second greatest action value at a given state, where $u_i^s \neq u_i^*, \forall i \in A$. Therefore, joint $Q$-values whose targets are smaller than $Q_{tot}(\boldsymbol{\tau}, \mathbf{u}^s)$ can be perfectly represented by QPLEX.

However, when $(\boldsymbol{\tau}, \mathbf{u}^*)$ is sampled, QPLEX tries to minimising the following loss:

$$(\sum_{i=1}^{n} \lambda_i(\boldsymbol{\tau}, \mathbf{u}^*)[Q_i(\boldsymbol{\tau}, u_i^*) - V_i(\boldsymbol{\tau})] - [Q(\boldsymbol{\tau}, \mathbf{u}^*) - V_{tot}(\boldsymbol{\tau})])^2, \tag{31}$$

where $Q(\boldsymbol{\tau}, \mathbf{u}^*) - V_{tot}(\boldsymbol{\tau}) > 0$, $Q_i(\boldsymbol{\tau}, u_i^*) - V_i(\boldsymbol{\tau}) \leq 0$, and $\lambda_i(\boldsymbol{\tau}, \mathbf{u}^*) > 0$. As a result, $\lambda_i(\boldsymbol{\tau}, \mathbf{u}^*)$ decreases to almost zero, and $Q_i(\boldsymbol{\tau}, u_i^*)$ gradually reaches to $V_i(\boldsymbol{\tau})$. However, the process is very time consuming and $Q_i(\boldsymbol{\tau}, u_i^*)$ cannot exceed $V_i(\boldsymbol{\tau})$ in most cases due to the low weight $\lambda_i(\boldsymbol{\tau}, \mathbf{u}^*)$, i.e., the **vanishing gradient** problem. Consequently, as shown in Figure 9h, the approximated joint $Q$-values for the optimal $\mathbf{u}^* = (u_1^1, u_2^1)$ and three suboptimal $\{(u_1^2, u_2^3), (u_1^3, u_2^2)(u_1^3, u_2^3)\}$ are 4.37 in QPLEX.

GVR uses inferior target shaping and superior experience replay to eliminate the non-optimal self-transition nodes. However, it ignores the stochasticity at all as WQMIX, and thus behaves as CW-QMIX and cannot converge to any joint action (Figure 9f). More comparison among GVR, WQMIX, and UTS can be found in Appendix D.

## C  ABLATION STUDIES

In this section, we conduct ablation studies to investigate the influence of target shaping in UTS. The following methods are included in the evaluation: 1) QMIX, which is the natural ablation baseline of UTS, 2) UTS without target shaping (denoted by UTS-wo-TS), i.e., agents are trained only according to the best action value function $q_a(s, \tau_a, u_a)$, 3) UTS without the reward predictor (denoted by UTS-wo-R), 4) UTS without the state predictor (denoted by UTS-wo-S), and 5) the original UTS.

### C.1  THE REWARD AND THE STATE PREDICTORS

First, we investigate the role of the reward and the state predictors. We compare UTS-wo-R, UTS-wo-S, and UTS on matrix game (stochastic non-monotonic rewards), multi-agent Markov Decision Process (MMDP) (stochastic state transition and non-monotonic returns), predator-prey (deterministic non-monotonic returns), and predator-stag-hare (stochastic non-monotonic returns). As shown in Figure 10, UTS-wo-R cannot cope with stochastic rewards and thus fail to solve matrix game, MMDP, and predator-stag-hare. UTS-wo-S cannot identify whether the state-action pair leads to stochastic transition and fail to solve MMDP. However, since predator-prey only involves deterministic targets, the target shaping function does not overestimate the suboptimal. As a result, both UTS-wo-R and UTS-wo-S can solve predator-prey, indicating the effectiveness of the target shaping function.

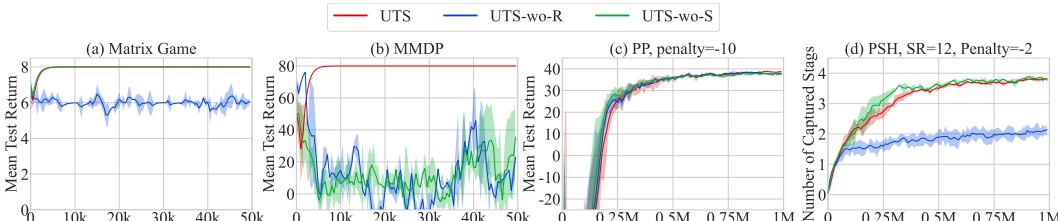

Figure 10: The comparison of UTS-wo-R, UTS-wo-S, and UTS.

The additional models (the reward and the state predictors) in UTS are introduced to identify stochasticity, which are necessary to cope with any target state-action values. When the task does not involve stochasticity, UTS learns slower because the target shaping function relies on the results from these models. When we have complete domain knowledge of the task, a suitable threshold for the model prediction error can be set to improve the learning speed.

### C.2  THE TARGET SHAPING FUNCTION

Now we show the effectiveness of the target shaping function. We compare QMIX, UTS-wo-TS, and UTS on predator-prey and predator-stag-hare. As shown in Figure 11, since the miscoordination penalty is large, QMIX cannot deal with non-monotonic targets and thus fail to solve these tasks.

UTS-wo-TS and UTS can learn to catch the prey (stags), whereas UTS learns more quickly. Since the estimated best action value introduces additional estimation error that may cause a significant variance, we also show the standard deviation of per-agent action values $Q_a$. As shown in Figure 11, we can see that UTS has a lower standard deviation than UTS-wo-TS.

To better understand why UTS can achieve better performance, first, we discuss the advantage of value factorisation methods. Linear value factorisation implicitly achieves credit assignment using a counterfactual baseline (Wang et al., 2021). The individual $Q$-value function $Q_i(s, u_i)$ is updated by:

$$\mathbb{E}_{u'_{-i} \sim \boldsymbol{\pi}}[y(s, u_i, u_{-i})] - \frac{n-1}{n} \mathbb{E}_{\mathbf{u}' \sim \boldsymbol{\pi}}[y(s, \mathbf{u}')] + w_i(s), \tag{32}$$

where the residue term $\mathbf{w} \equiv [w_1(s), ..., w_n(s)]$ denotes an arbitrary vector satisfying $\forall s, \sum_1^n w_i(s) = 0$. As a result, the individual $Q$-value in linear value factorisation can be viewed as a type of *advantage function*, where the per-agent baseline reduces variance but does not affect the convergence. Due to the non-linear mixing function in QMIX, a more general theoretical understanding of monotonic value factorisation is still insufficient. However, previous work shows that QMIX also implicitly performs credit assignment and outperforms linear value factorisation (Rashid et al., 2018; 2020; Zhou et al., 2020).

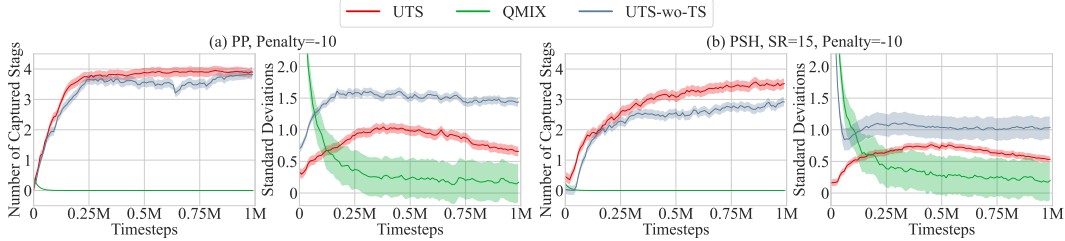

Figure 11: The comparison of QMIX, UTS, and UTS-wo-TS.

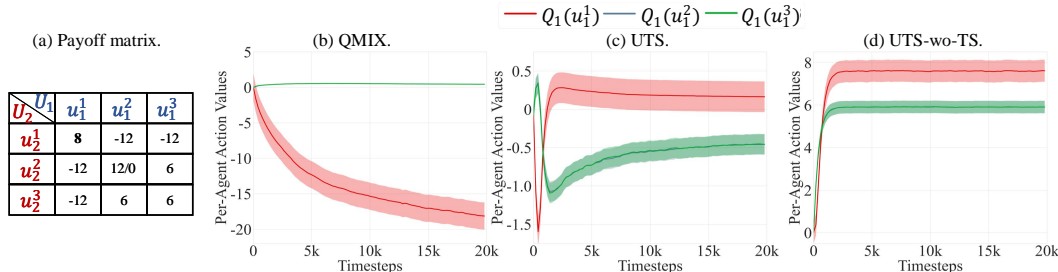

Figure 12: Per-agent action values from QMIX, UTS, and UTS-wo-TS.

A numerical example is provided in Figure 12. Figure 12a shows the payoff of a one-step non-monotonic game. First, we show the standard deviation of per-agent action values $Q_a$. QMIX converges to the suboptimal when it meets non-monotonic targets of the joint $Q$-value function. As shown in Figure 12b, $Q_1^{QMIX}(u_1^1)$ keeps reducing and is smaller than the values of other actions, which implies that QMIX converges to the suboptimal due to representational limitation. Both UTS and UTS-wo-TS can recover the optimal policy. The action values in UTS-wo-TS show similar trends, and thus UTS-wo-TS can be stable until it perfectly approximates all action values. By contrast, UTS shapes the targets to ensure they can be represented by monotonic value factorisation, where the optimal policy is invariant during this shaping. UTS performs credit assignment with the shaped targets, the action values in UTS can be viewed as a type of *advantage* and thus are much smaller than UTS-wo-TS. The action values in

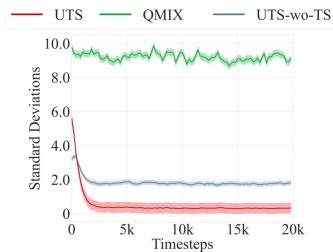

Figure 13: Standard deviation of per-agent action values from QMIX, UTS, and UTS-wo-TS on a random $10 \times 10$ matrix game.

UTS show two different trends, where $Q_1^{UTS}(u_1^1)$ is positive, $Q_1^{UTS}(u_1^2)$ and $Q_1^{UTS}(u_1^3)$ are negative when the algorithm converges.

We also show the standard deviation of per-agent action values from QMIX, UTS, and UTS-wo-TS on a random $10 \times 10$ matrix game, where the suboptimal is filled with random numbers generated uniformly between -20 and 19, and the unique optimal value is +20. As shown in Figure 13, UTS reduces variance compared to UTS-wo-TS, thus improving learning efficiency and stability.

## D   THE COMPARISON WITH WQMIX AND GVR

In Section 3.1, we prove that WQMIX with a low $\alpha$ can overcome any non-monotonic tasks. However, we also prove that WQMIX is particularly brittle with stochasticity because it overestimates the expected value of the suboptimal whose target is large with a low probability. GVR also ignores the stochasticity and cannot accurately identify whether a given action is inferior. As a result, WQMIX with fine-tuned $\alpha$ and GVR are able to solve any non-monotonic targets, but cannot deal with stochasticity.

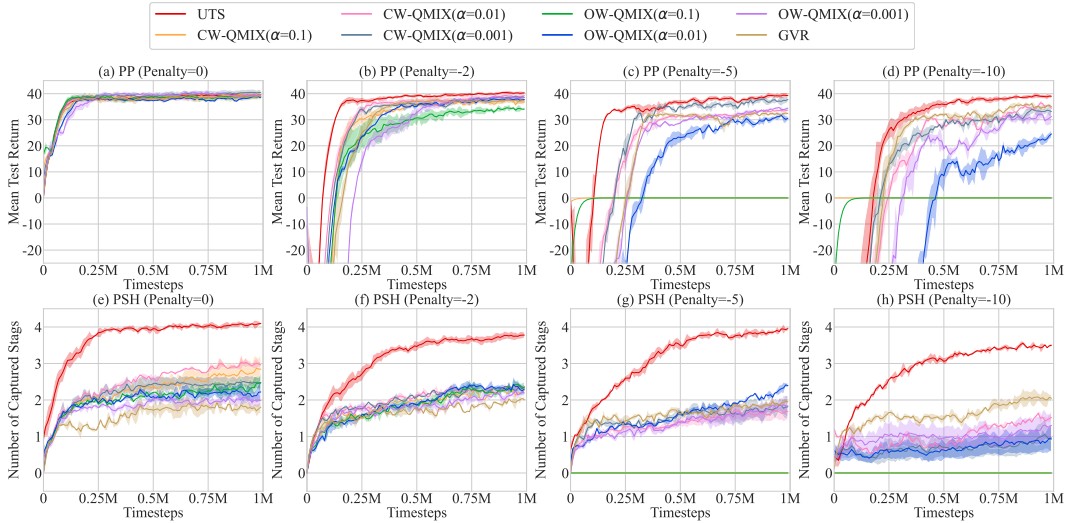

Figure 14: The comparison of GVR, UTS, and WQMIX with different weights.

To further show the difference between these methods and UTS, we compare them on predator-prey with different level of miscoordination penalties and stochasticity. Figure 14 demonstrates that WQMIX with a very low $\alpha$ and GVR can obtain the positive test return in all penalty settings on predator-prey. However, they cannot recover the optimal joint policy on stochastic predator-stag-hare. By contrast, UTS outperforms WQMIX and GVR in these tasks.

## E   THE COMPARISON WITH MORE ALGORITHMS

We also compare our method with more algorithms to further show our motivation. We consider recent multi-agent reinforcement learning works, including RODE (Wang et al., 2020c), MAPPO (Yu et al., 2021), and HAPPO (Kuba et al., 2021).

Figure 15a-d shows that RODE, MAPPO, and HAPPO fail to recover the optimal policy in the matrix game and the MMDP with stochastic and non-monotonic targets, where the payoff is shown in Section 6.1. In addition, they also cannot solve predator-prey and predator-stag-hare when the penalty is large. This is because that RODE applies the same monotonic mixing network as QMIX and thus can only represent values in the restricted monotonic space. This representational limitation makes RODE cannot cope with non-monotonic targets. MAPPO is an independent learning approach with many valuable techniques, but it cannot deal with these tasks due to the non-stationarity of the environment. HAPPO applies an agent-by-agent update scheme and improves the performance

compared to MAPPO. This update scheme can only guarantee convergence to one of the Nash Equilibriums in theory (each agent is treated to receive the equal reward under the team reward setting). However, the monotonic target in value decomposition methods can be interpreted as a static multivariable optimisation problem, where there are non-optimal solutions that cannot be improved upon by coordinate descent. The sequential policy update scheme cannot guarantee convergence to the optimal in this optimisation problem (Bertsekas, 2019). Or we can also view the task of predator-prey and predator-stag-hare as a Pareto Selection problem, and HAPPO cannot ensure convergence to the real optimal (the Pareto optimal) and thus fail to solve these tasks.

The representational limitation of monotonic value decomposition will not necessarily result in converging to suboptimal. We can see that RODE outperforms UTS in some SMAC maps. However, UTS still achieves comparable performance with RODE in many SMAC tasks, indicating that the target shaping function is more general than the weighting function in WQMIX.

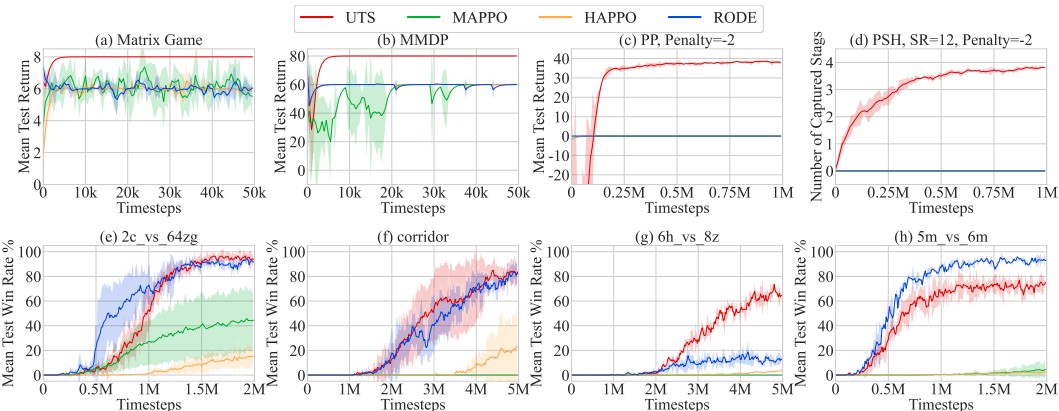

Figure 15: The performance of UTS, RODE, MAPPO, and HAPPO on matrix game, MMDP, predator-prey, predator-stag-hare, and many SMAC maps.

The sampling efficiency of MAPPO and HAPPO is lower than $Q$-learning based algorithms because they are on-policy algorithms. Therefore, Figure 15e-h shows their poor performance compared with RODE and UTS with limited environment steps. Not that we use `SC2.4.6.2.69232` (the version of StarCraft II which VDN, QMIX, and RODE used in their papers) instead of `SC2.4.10` (the version which policy-based algorithm always used). Performance is not always comparable between versions.

We emphasise that our work is to provide a concrete analysis of the relationship between the non-monotonic and stochastic targets and the monotonic value decomposition. We propose a target shaping function to eliminate the adverse impact from representational limitation, where the approximation of all original targets is not required. The shaping function project all original targets to a monotonic space and keeps the optimal policy unchanged. Since all shaped targets are tractable to monotonic value decomposition, QMIX can recover the optimal policy with any original target. That is to say, we relax the monotonic assumption of the relationship between the joint $Q$-value and individual $Q$-values.

## F    EXPERIMENTAL SETUP

We adopt the same architecture for VDN, QMIX, QTRAN, QPLEX, and WQMIX as (Samvelyan et al., 2019; Rashid et al., 2020). Each agent independently learns a policy with fully shared parameters between all policies. We used RMSProp with a learning rate of $5 \times 10^{-4}$ and $\gamma = 0.99$, buffer size 5000, mini-batch size 32 for all algorithms. We use an $\epsilon$-greedy strategy, and $\epsilon$ is linearly annealed from 1 to 0.05 over $50k$ timesteps (unless specified otherwise). The dimension of each agent's GRU hidden state is set to 64.

For WQMIX, we use $\alpha = 0.1$ on predator-prey and predator-stag-hare (unless specified otherwise), and $\alpha = 0.5$ in SMAC.

For UTS, we use the same agent and mixing network in WQMIX as (Rashid et al., 2020). For the reward and the state predictors, we use a multilayer perceptron (MLP) with ReLU non-linearities and a gated recurrent unit (GRU) to extract features, where the hidden dim is 128. Then we use another MLP to output the predicted reward, the standard deviation and the embedding of the next state. The first and the GRU parameters are shared. The dimension of the target embedding for the state $m$ is set to 16. We use Adam with a learning rate of $1 \times 10^{-3}$ to train the predictors. For the best $Q$-value function, we use feed-forward networks with three hidden layers of dim $\{64, 64\}$ and ReLU non-linearities, where the optimiser is RMSProp with a learning rate of $5 \times 10^{-4}$.

We conduct experiments on an NVIDIA RTX 3090 GPU. Each task needs to train for about 5 hours on predator-prey, and about 8 to 16 hours on SMAC, depending on the number of agents and episode length limit of each map. We evaluate 32 episodes with decentralised greedy action selection every $10k$ timesteps for each algorithm.

We use an $\epsilon$-greedy policy in which $\epsilon$ decreases from 1 to 0.05 over $1M$ timesteps for *3s5z_vs_3s6z*, *6h_vs_8z* and *corridor*, and over $50k$ timesteps for others.

All the learning curves in the experiments are plotted based on five training runs with different random seeds using mean and standard deviation with confidence internal 95%.

