# OpenReview forum: "UTS: When Monotonic Value Factorisation Meets Non-monotonic and Stochastic Targets"
_ICLR.cc/2023/Conference — Submitted to ICLR 2023_

### Official Review · Reviewer_9h2R · 2022-10-20

**Confidence:** 4
**Correctness:** 2
**Technical Novelty And Significance:** 2
**Empirical Novelty And Significance:** 2
**Recommendation:** 3

**Clarity, Quality, Novelty And Reproducibility:**

I personally found some aspects of the clarity to be lacking, as discussed above. I also raised some issues regarding the quality of the empirical approach (ie, dealing with stochasticity) and the quality of the baselines and benchmarking. To my knowledge, the approach is novel. The code is not available; I would guess that it would difficult to independently reproduce the results.

**Strength And Weaknesses:**

### Clarity

I personally found the description of the problems that UTS is trying to solve somewhat obfuscated. Perhaps I am not the target audience and that is why I found it difficult to follow, but I think the submission could be clearer overall. To give a concrete example, I feel like I read through the whole intro without really understanding the main idea.

### Baselines

It seems like the submission is lacking some relevant baselines, such as MAPPO, IPPO, and HATRPO, given that it is claiming to "outperform state-of-the-art".

### Benchmarking

In deep RL, and perhaps especially deep MARL, there is historical precedent for bad empirical practices, such as reporting mean return over a small number of seeds. It is well documented that this practice leads to unreliable reporting of algorithmic strength -- see Deep Reinforcement Learning at the Edge of the Statistical Precipice (NeurIPS 2021). I think the submission ought report the alternative metrics advocated in the statistical precipice paper if it wants to make big claims like "outperforming state-of-the-art".

### Motivation

> In this section, we introduce a shaping function to generate the shaped joint action-value Qf(s, u) to replace the original Q(s, u). The shaping function should have the following properties: 1) policy invariance, i.e., not making the agent deviate from the true goal and keeping the optimal policy unchanged; 2) full representational capacity, i.e., all shaped joint action-values should belong to
a subset of Qmvf.

This is a really important sentence in the submission. Yet, it doesn't feel clear to me. What does "not making the agent deviate from the true goal" mean? Additionally, why are we saying that "shaped joint action-values should belong to a subset of Qmvf"? Isn't belonging to a subset of a set the same as belonging to the set itself?

### Dealing with stochasticity

The approach to determining whether stochasticity is present feels unsatisfactory to me. The state predictor may output zero even if there is stochasticity.

### Confusion

> However, these approaches suffer from instability arising from the non-stationarity of the environment induced by simultaneously
learning and exploring agents (Kuba et al., 2021). Centralised learning of joint action-values can solve these problems, but it is challenging to scale in real-world applications due to intractably large joint action space and communication constraints.

Kuba et al. (2021) show that MAPPO-style approaches lack a monotonic improvement guarantee. Unless I am misunderstanding, UTS does not have such a guarantee either. Could the authors clarify this point?

**Summary Of The Paper:**

The submission proposes a new CTDE algorithm called UTS. UTS is designed to have two properties that the submission calls policy invariance and full representational capacity. The submission benchmarks UTS in matrix games, predator prey, and SMAC.

**Summary Of The Review:**

The method feels a little hacky to me, but I think that's not the worst thing if it performs well.

The submission makes big claims about its method's performance, such as "outperforming state-of-the-art". However, I do not feel that it lives up to these claims in the experiments, both because it is lacking in baselines and because it does not follow modern deep RL benchmarking practices.

---

> ### Author Response · Authors · 2022-11-17
> **Response for 9h2R**
>
> > Clarity.
>
> Thanks for pointing this out! We have revised the introduction to make the overall problem become clearer. We also briefly introduce it as follows:
>
> **a**. Decentralized learning suffers from non-stationarity. Centralized learning is the only way to guarantee to achieve global optimal in the MARL setting. However, centralized execution is not acceptable when communication is restricted. Centralized learning with a decentralized execution (CTDE) framework can solve this problem. One of the most significant challenges for these methods is keeping the optimal consistency between centralized policy ($Q_{tot}$) and decentralized policies ($Q_i$), i.e., the Individual-Global-Max (IGM) principle.
>
> **b**. Value decomposition methods are proposed to extract the decentralized policies from the centralized joint action value. Monotonic value decomposition is the easiest way to satisfy the IGM principle and shows good performance on SMAC tasks.
>
> **c**. However, monotonic value decomposition can only represent monotonic target value functions $Q$ (the representational limitation). Therefore, the optimal joint action in the approximated $Q_{tot}$ may not be the same as the real optimal, $\arg\max Q_{tot} \neq \arg\max Q$. Clearly, the decentralized policies $Q_i$ from $Q_{tot}$ is not the real optimal. Many recent works have pointed out this problem.
>
> **d**. There are two possible solutions to solve the representational limitation. First, we can introduce the joint action into the input of the centralized value function and improve its representational capacity. However, it is difficult to learn such centralized value functions due to the large state-action space. Second, we can prioritize the optimal joint action to learn biased centralized value functions, i.e., focusing on the representation of the optimal value rather than the suboptimal. Weighted QMIX (WQMIX) and our method belong to the latter one.
>
> **e**. However, we analyze the fundamental limitation of the weighting function in WQMIX and show it cannot cope with non-monotonic and stochastic target value functions. We propose an alternate way to solve this problem. Since QMIX cannot represent all values in $Q$, we can use a shaping function to project $Q$ into $Q^f$ (lies in the monotonic space) and ensures that $\arg\max Q=\arg\max Q^f$. Then QMIX can represent all values in $Q^f$, and then converge to the real optimal w.r.t both $Q^f$ and $Q$.
>
> > Baselines and benchmarking.
>
> Yes, the recent trend in cooperative MARL is that well-tuned baselines perform competitively with more complicated approaches. However, these baselines are far away from our motivation, and thus we do not include them in our comparison.
>
> Motivation: On the one hand, we focus on the optimal consistency between the centralized policy and decentralized policies, which is one of the most significant challenges in value decomposition methods under CTDE settings. MAPPO and IPPO do not involve the concept of centralized policy. On the other hand, we consider the non-monotonic and stochastic target value functions. MAPPO, IPPO and HATRPO/HAPPO cannot cope with non-monotonic targets (which can be interpreted as the relative overgeneralization pathology). MAPPO and IPPO apply independent learning methods and thus converge to the suboptimal. HATRPO/HAPPO applies an agent-by-agent update scheme and improves the performance compared to MAPPO. However, the monotonic target in value decomposition methods can be interpreted as a static multivariable optimisation problem, where there are non-optimal solutions that cannot be improved upon by coordinate descent. The sequential policy update scheme cannot guarantee convergence to the optimal in this optimisation problem[1]. Or we can also view the task of predator-prey and predator-stag-hare as a Pareto Selection problem, and HAPPO cannot ensure convergence to the real optimal (the Pareto optimal) and thus fail to solve these tasks.
>
> Experiments: We have to point out that the performance is not always comparable between different versions of SMAC. MAPPO achieves 100\% win rates in almost all SMAC maps because it uses the 4.10 version rather than 4.6 (which is used in VDN, QMIX, QPLEX, etc.). On-policy based algorithms always use more environment steps than QMIX-based algorithms.
>
> We also compare these methods, and the results can be found in the new revision (Appendix E).

---

> > ### Author Response · Authors · 2022-11-17
> > **Response for 9h2R (Part II)**
> >
> > > Motivation.
> >
> > Thanks for pointing this out. We have modified this sentence in the revision to make it become clearer.
> > In monotonic value decomposition, the approximate joint action value $Q_{tot}$ cannot represent target value functions $Q$ which are non-monotonic. It is possible to achieve full representational capacity w.r.t $Q$ by introducing joint action into the input for $Q_{tot}$, but it is expensive because the joint action space is intractably large. Therefore, we introduce a shaping function to shape the original targets $Q$ to the shaped targets $Q^f$. The shaping function satisfies policy invariance ($\arg\max Q^f = \arg\max Q$) and full representational capacity w.r.t $Q^f$ rather than $Q$. The advantage is that $Q^f$ is monotonic and satisfies the assumption of monotonic value decomposition (the target value functions should be monotonic), and thus we can represent all $Q^f$ and converge to the optimal (w.r.t both $Q^f$ and $Q$).
> >
> > > Dealing with stochasticity.
> >
> > The idea of the fixed and randomly initialised target network for the state prediction problem stems from RND[2], a commonly used exploration method in RL. Many works share the same idea with this method, e.g., the autoencoder in anomaly detection.
> >
> > If the state-action pair has never been visited, the state and the reward predictor cannot correctly identify whether its leads to stochasticity. In this case, our method degenerates to QMIX with target shaping (replacing every suboptimal by the best action value) and thus can only cope with non-monotonic targets (as WQMIX). It requires more visitations to solve stochasticity.
> >
> > > Confusion.
> >
> > The fundamental idea of MAPPO-based and QMIX-based algorithms is different.
> >
> > Although MAPPO is developed under the CTDE setting, it uses state information rather than observation as the value function $V(s)$ input (with many practical and useful tricks). The update rule for the policy network is the same as independent learning. However, independent learning suffers from non-stationarity and thus may not converge in some MARL tasks. That is why HAPPO introduces other agents’ actions into the learning procedure.
> >
> > However, value decomposition methods focus on extracting decentralized policies ($Q_i$) from centralized learning ($Q_{tot}$). The joint action value function $Q_{tot}=f(Q_i)$ is a special concept in value decomposition methods, where $f$ is the mixing network. $Q_i$ is trained by directly minimizing the error between $Q_{tot}$ and the target $y$. Therefore, the most important thing is to guarantee the optimal consistency between $Q_{tot}$ and $Q_i$, i.e., the Individual-Global-Max (IGM) principle. Monotonic value decomposition is one of the easiest ways to achieve IGM. However, it also leads to representational limitation ($Q_{tot}$ cannot represent the nonmonotonic target value functions Q). Therefore, $\arg\max Q_{tot}\neq \arg\max Q$.
> >
> > There are two possible solutions to solve the representational limitation. First, we can introduce the joint action into the input of the centralized value function and improve its representational capacity. However, learning such centralized value functions is difficult due to the large state-action space. Second, we can prioritize the optimal joint action to learn biased centralized value functions, i.e., focusing on the representation of the optimal value rather than the suboptimal. Weighted QMIX (WQMIX) and our method belong to the latter one.
> > In UTS, we shape the original targets $Q$ to $Q^f$, which is monotonic and can be represented by monotonic value decomposition. In addition, we guarantee that $\arg\max Q=\arg\max Q^f$. Then, QMIX can achieve the full representational capacity w.r.t $Q^f$ rather than $Q$ and converge to the real optimal $\arg\max Q^f=\arg\max Q$.
> >
> > [1] Dimitri Bertsekas. Multiagent rollout algorithms and reinforcement learning. arXiv preprint arXiv:1910.00120, 2019.
> >
> > [2] Yuri Burda, Harrison Edwards, Amos Storkey, and Oleg Klimov. Exploration by random network distillation. arXiv preprint arXiv:1810.12894, 2018.

---

### Official Review · Reviewer_gapB · 2022-10-24

**Confidence:** 5
**Correctness:** 2
**Technical Novelty And Significance:** 2
**Empirical Novelty And Significance:** 2
**Recommendation:** 3

**Clarity, Quality, Novelty And Reproducibility:**

- Writing needs improvement as the paper is not very clear. Related works also needs to address MARL approaches more broadly.
- The paper tries to fix problems in an existing method.
- There is also need for better empirical evaluation on complex domains with actual stochasticity. The performance compared to existing methods like [3] is also not well. Some of the claims about representability achieved by reward shaping is not well supported with proofs.
- SMAC is an opensourced domain, the  code for the method and predator prey environment has not been provided.

**Strength And Weaknesses:**

1. There are some important works missing in the related works which need to be discussed towards getting a proper context for learning in CTDE settings (instead of focusing only on value based methods) eg. : Tesseract [1] an actor-critic method natively supports decentralization despite environment stochasticity and non monotonicity, it also proves important representation capacity results. Similarly, there have been developments in model based methods [2] for addressing representational issues.

2. How does the proposed method deal with uncertainty arising from neural network approximation? this is a huge component of target noise in practise. There isn't enough discussion on this given that the proposed method relies on uncertainty estimates that too with limited number of environment steps.

3. What is meant by "convergence to the optimal QMIX for any non-monotonic and stochastic Q"? is it the best qmix learnable policy?

4. How exactly is eq 7 preventing a bad monotonic projection? There needs to be a guarantee in probabilisitic or worst case sense.

5. Why are stronger baselines not used for comparison? For eg. [3] has better performance on all the SMAC maps considered here. Overall this also affects significance of the work if strictly better methods already exist.

6. The paper claims to be aimed for stochastic domains but SMAC is not adequately stochastic.

7. The paper needs improvement in terms of writing and presentation, especially for conveying the overall problem and method. Currently, I do not find it easily readable.

Minors:
1. The results used from other works should be cited where they are used: eg. The line in introduction "However, QMIX can only represent values in the restricted monotonic space" should cite Maven, Mahajan et al 2019 and Qtran, Son et al 2019, there itself.
2. Eq 3 has been repeated twice.
3. Pg.5 ending, "objection of", did you mean "objective of"?

References:
1. Tesseract: Tensorised Actors for Multi-Agent Reinforcement Learning, Mahajan et al, 2021
2. Model based Multi-agent Reinforcement Learning with Tensor Decompositions, Van Der Vaart et al 2021
3. Rode: Learning roles to decompose multi-agent tasks, Wang et al, 2021


**Summary Of The Paper:**

This paper proposes a reward shaping scheme for MARL towards fixing QMIX, an existing algorithm suffering from suboptimality in non-monotonic settings. The method uses target shaping by predicting reward and next state uncertainty.

**Summary Of The Review:**

Interesting idea but needs improvements (see the 2 sections above).

---

> ### Author Response · Authors · 2022-11-17
> **Response for Gap8 (Part I)**
>
> Thanks for your comments!
>
> > There are some important works missing in the related works which need to be discussed towards getting a proper context for learning in CTDE settings.
>
> Thanks for providing these interesting papers! We have added more CTDE-based algorithms in the revision (please refer to the related work section), including MAPPO[1], HATRPO/HAPPO[2], RODE[3], Tesseract[4], and Deep Coordination Graph (DCG)[5].
>
> As we all know, monotonic value factorisation (MVF) ensures the optimal consistency between the approximated optimal joint policy ($\arg\max \hat{Q_{tot}}$) and the individual optimal ($\arg\max \hat{Q_{i}}$), i.e., Individual-Global Max. However, due to the representational limitation, there is a gap between the approximated action value $\hat{Q_{tot}}$ and the real target $Q$.
>
> Some works try to achieve full representational capacity (reduce the gap between $\hat{Q_{tot}}$ and $Q$) by introducing the joint action (or pairwise interactions) as the input for $\hat{Q_{tot}}$, e.g., DCG, Tesseract, and QPLEX. However, approximating such joint action value is laborious because the joint action space grows with the number of agents.
>
> Another solution is to prioritise the optimal action and learn a biased $\hat{Q_{tot}}$, e.g., Weighted QMIX and EMC. These methods do not achieve full representational capacity and only try to make $\arg\max \hat{Q_{tot}} = \arg\max Q$, which is the goal of the MARL algorithms. Our method belongs to this group. We do not need to approximate each $Q$. The shaping function projects any $Q$ to the monotonic space and generates shaped targets $Q^f$, where $\arg\max Q^f = \arg\max Q$. Therefore, QMIX can achieve the full representational capacity w.r.t the shaped targets $Q^f$ and converge to $\arg\max Q^f$, which is also optimal for the original optimal for $Q$.
>
> > How does the proposed method deal with uncertainty arising from neural network approximation?
>
> The uncertainty estimation in our method is inspired by the Bayesian deep learning method and random network distillation[6].
>
> The noise of the model (reward and the state predictors) can be attributed to many factors, e.g., the amount of training data, stochastic target function, model misspecification, and learning dynamics[6]. With the limited number of environment steps (insufficient data due to exploration), the reward and the state predictors cannot correctly identify whether the state-action pair leads to stochasticity. In this case, our method degenerates to QMIX with target shaping, which can only cope with non-monotonic targets. It requires more visitations to solve stochasticity.
>
> In this paper, we focus on the representation of joint action value rather than the uncertainty arising from neural network approximation (e.g., we assume that the model is accurate in theoretical results). It is possible to analyse when we can trust our model (Model-based policy optimisation [7]), which is a promising way and we can investigate in future work.
>
> > What is meant by "convergence to the optimal QMIX for any non-monotonic and stochastic Q"?
>
> Thanks for pointing this typo out. We are trying to say that the shaping function can shape the target value function to be monotonic for a given MDP with random reward functions (i.e., non-monotonic and stochastic). QMIX achieves full representational capacity w.r.t the shaped targets and thus can converge to the real optimal.
>
> > How exactly is eq 7 preventing a bad monotonic projection?
>
> Please refer to Theorem 2 for more details. We prove that we can find a monotonic function representing all shaped values.
>
> > Why are stronger baselines not used for comparison?
>
> Because our motivation is different from these methods. Recently, many QMIX-based algorithms have been proposed to improve coordination from many perspectives, e.g., role-based learning, multi-agent exploration, etc. However, since these methods apply the same non-negative weights in the mixing network, they can represent the exact same class of joint action values as QMIX. Therefore, they still suffer from the limited representational capacity problem (i.e., non-monotonic and stochastic target value functions) introduced by monotonic value factorisation, which is our paper’s starting point.
>
> We also compare additional baselines, including RODE, MAPPO, and HAPPO, and the results can be found in the revision (Appendix E). Neither of these methods can cope with non-monotonic and stochastic targets.

---

> > ### Author Response · Authors · 2022-11-17
> > **Response for Gap8 (Part II)**
> >
> > > The paper claims to be aimed at stochastic domains but SMAC is not adequately stochastic.
> >
> > We conduct experiments on the matrix game, multi-agent Markov decision process, predator-prey, and predator-stag-hare, which have considerable stochasticity (reward and state transition) and non-monotonic target value functions.
> >
> > We use SMAC as a benchmark because we believe the proposed algorithm should be applicable for more general tasks (rather than the environment with non-monotonic and stochastic targets only). For example, WQMIX uses a weighting function to help QMIX solve non-monotonic targets, and it can cope with predator-prey tasks. However, it obtains low performance on many SMAC maps (compared with QMIX). By contrast, our method achieves comparable performance on many SMAC maps.
> >
> > > Writing (Eq 3 has been repeated twice and typos) and presentation.
> >
> > We have revised our paper (especially the introduction, related work, and experiments) to clarify the overall problem.
> >
> > Eq.3 is the WQMIX operator ($Q_{tot}\in Q^{mvf}$ with $w(s,u)$). In the Background, we introduce the QMIX operator ($Q_{tot}\in Q^{mvf}$ without $w(s,u)$). In the Case Studies, we introduce the WVDN operator ($Q_{tot}\in Q^{lvf}$). They are different operators.
> >
> > [1] Chao Yu, Akash Velu, Eugene Vinitsky, Yu Wang, Alexandre Bayen, and Yi Wu. The surprising effectiveness of PPO in cooperative, multi-agent games. arXiv preprint arXiv:2103.01955, 2021.
> >
> > [2] Jakub Grudzien Kuba, Ruiqing Chen, Munning Wen, Ying Wen, Fanglei Sun, Jun Wang, and Yaodong Yang. Trust region policy optimisation in multi-agent reinforcement learning. arXiv preprint arXiv:2109.11251, 2021.
> >
> > [3] Tonghan Wang, Tarun Gupta, Anuj Mahajan, Bei Peng, Shimon Whiteson, and Chongjie Zhang. Rode: Learning roles to decompose multi-agent tasks. arXiv preprint arXiv:2010.01523, 2020.
> >
> > [4] Anuj Mahajan, Mikayel Samvelyan, Lei Mao, Viktor Makoviychuk, Animesh Garg, Jean Kossaifi, Shimon Whiteson, Yuke Zhu, and Animashree Anandkumar. Tesseract: Tensorised actors for multi-agent reinforcement learning. In International Conference on Machine Learning, pp. 7301–7312. PMLR, 2021.
> >
> > [5] Wendelin B¨ohmer, Vitaly Kurin, and Shimon Whiteson. Deep coordination graphs. In International Conference on Machine Learning, pp. 980–991. PMLR, 2020.
> >
> > [6] Yuri Burda, Harrison Edwards, Amos Storkey, and Oleg Klimov. Exploration by random network distillation. arXiv preprint arXiv:1810.12894, 2018.
> >
> > [7] Janner, M., Fu, J., Zhang, M., & Levine, S. (2019). When to trust your model: Model-based policy optimisation. Advances in Neural Information Processing Systems, 32.

---

### Official Review · Reviewer_kwZB · 2022-10-27

**Confidence:** 4
**Correctness:** 3
**Technical Novelty And Significance:** 3
**Empirical Novelty And Significance:** Not applicable
**Recommendation:** 6

**Clarity, Quality, Novelty And Reproducibility:**

The use of random embedding for estimating stochasticity seems similar to some work on exploration such as the following (although with a different goal).  Is there a relation?

@article{burda2018exploration,
  title={Exploration by random network distillation},
  author={Burda, Yuri and Edwards, Harrison and Storkey, Amos and Klimov, Oleg},
  journal={arXiv preprint arXiv:1810.12894},
  year={2018}
}

The first sentence of 4.2 is confusing to me.  I thought Q^{mvf} is intended to be the set of things that QMIX can in principle learn.  But the phrasing suggests that there are things in it that QMIX cannot represent.  Is there a typo here or something that needs to be better explained?

The reason of predicting each agent’s best action value function is not well explained in the main text and required studying the proof of Theorem 2 for me to understand.  My intuition is the goal is quantify the suboptimality of actions because, by definition, the the join action is suboptimal (and the optimal action is unique) then for every agent not playing their part of the optimal action their q_a will be less than the optimal Q value.

The statement of Theorem 2 seems to be missing something.  It contains a “such that,” with no indication what the assumption is.  From the proof I’m guessing it is that the optimal policy is unique and the discussion of this issue in the proof should be discussion and not actually part of the proof?  Relatedly, I don’t know where there is a quantification over joint actions u in the theorem statement.

The intuition for the decision not to change the target if it is highly stochastic is not clear to me. (Discussion before (8))


**Strength And Weaknesses:**

Strengths
•	Incorporates stochasticity in reward and state representation
•	Good empirical performance
•	Limitations of prior works is explained through theory and the example of matrix game

Weaknesses
•	The overall approach is a bit of a kitchen sink with many hyperparameters to be tuned and models to be trained. QMIX alone is already known for being challenging on this front with larger number of agents.  See, e.g., the below reference.
•	Goal of estimating the stochasticity of rewards and next states seems similar in spirit to distributional RL, but there is no comparison to related work from this literature.
•	While the experiments include some ablations, the role of various parts of the overall approach is not fully explored.  For example, there are two approaches used in combination to detect stochasticity (reward predictor and state predictor).  What if you only include one?  Or what happens if you use different rules about when to apply or not apply the change of target than that in Equation (7) since at the very least it has two different cases.

@article{avalos2021local,
  title={Local Advantage Networks for Cooperative Multi-Agent Reinforcement Learning},
  author={Avalos, Rapha{\"e}l and Reymond, Mathieu and Now{\'e}, Ann and Roijers, Diederik M},
  journal={arXiv preprint arXiv:2112.12458},
  year={2021}
}


**Summary Of The Paper:**

The paper proposes a new value factorization method for cooperative MARL. The technique brings out an alternate way of value factorization (to WQMIX and QMIX) when the target values are non-monotonic and stochastic. In particular, prior approaches is the inability to recover the optimal Qtot value when the joint value function is non-monotonic and stochastic. A popular matrix game example variant shows that the previous approaches frequently overestimate or underestimate joint Q values in stochastic and non-monotonic settings. An alternate shaping function is introduced to generate the shaped joint action-value that can consider the stochasticity in the reward and state function and place appropriate weight on the different joint actions. To this end, a neural network is trained to predict standard deviation and error in the reward and the embedding of the next state. Another neural network is trained to estimate the best action value for each agent, assuming the agent gets the cooperation of other agents. Finally, based on the prediction error and the best action values, the shaping function generates monotonic target joint action-values for QMIX.

**Summary Of The Review:**

A paper with strong empirical performance but room for improvement in the throughness and clarity of the exposition.

---

> ### Author Response · Authors · 2022-11-17
> **Response for Reviewer kwZB**
>
> Thanks for your comments and insight!
>
> > The use of random embedding for estimating stochasticity seems similar to some work on exploration such as the following (although with a different goal). Is there a relationship?
>
> Yes. We got inspired by RND to design the fixed and randomly initialised target network for the state prediction problem. We have added this citation in the revision.
>
> > The first sentence of 4.2 is confusing to me. I thought Q^{mvf} is intended to be the set of things QMIX can learn in principle. But the phrasing suggests that there are things in it that QMIX cannot represent. Is there a typo here or something that needs to be better explained?
>
> We are trying to say that QMIX can only represent a subset of Q^{mvf} in practice due to the parametrisation of the mixing function and agent utilities. Since it may lead to misunderstanding, we have modified this paragraph in the revision.
>
> > The reason of predicting each agent’s best action-value function is not well explained in the main text and required studying the proof of Theorem 2 for me to understand.
>
> Thanks for your advice! We have revised this part to make it easy to follow.
>
> QMIX cannot represent the non-monotonic target $Q$-values (and it is unnecessary). We just need to represent the suboptimality of uncooperative actions rather than their exact values. Therefore, we propose the best action-value function to quantify the suboptimality and use this value to shape the original targets.
>
> > The intuition for the decision not to change the target if it is highly stochastic is not clear to me.
>
> Since QMIX places uniform weights on different joint actions, it can estimate the correct expected value of the stochastic target already. Therefore, we do not change the target if it is highly stochastic.
>
> Indeed, since the best action value function $q_i$ approximates the expected return when the target is stochastic, we can replace the target with $\frac{1}{n}\sum_{i} q_i$ rather than $\min q_i$ for the stochastic targets. This may improve the approximation in an ensemble fashion. However, considering that it introduces additional model errors (since $q_i$ is always used to approximate the best action value), we still use the original target.
>
> > The overall approach is a bit of a kitchen sink with many hyperparameters to be tuned and models to be trained.
>
> Yes. UTS involves more hyperparameters because it uses the model error to identify stochasticity, which draws inspiration from the multi-agent exploration method (RND). We also tried to use a visitation-based method (e.g., locality-sensitive hashing) to record the value or the embedding, which still requires a huge number of hyperparameters.
>
> With the development of the RL methods, the way to achieve uncertainty estimation and the best action-value function can be improved in the future.  In this paper, we mainly focus on the analysis of WQMIX in stochastic environments and an alternate way to eliminate the impact from representational limitation (i.e., the target shaping function, which makes QMIX achieve full representational capacity w.r.t the shaped targets rather than the original ones).
>
> > The presentation and the experiments.
>
> Thanks for going through our proofs! We have improved the presentation of the theory. In addition, we conduct more ablation studies to investigate the role of the reward predictor and the state predictor, and show more comparisons with MARL algorithms. The results can be found in Appendix C.1 and E in the revision.

---

### Decision · Program_Chairs · 2023-01-20

**Decision:**

Reject

**Justification For Why Not Higher Score:**

Some of the reviewers complained that the experimental evaluation was not thorough enough. This is partially something that the entire field of MARL suffers from but the authors should have at least compared to MAPPO in their original submission, rather than doing this only during the rebuttals.
The method it also rather complicated and it would be great to have a more in-depth ablation study of the different components.

On balance I believe it would be preferable to submit an updated manuscript which addresses all of the issues raised by the reviewers to a future conference for a fresh and unbiased set of reviews.

**Justification For Why Not Lower Score:**

N/A

**Metareview: Summary, Strengths And Weaknesses:**

Summary:
This paper aims to address two shortcomings of the popular methods QMIX for and WQMIX for learning in Dec-POMDPs.
Through simple toy example the authors show that these methods struggle both with non-monotonic and stochastic settings and propose a methods (UTS) that addresses these issues.

Strength:
I believe that there is real insight in this paper and that the analysis of failure cases / conceptual parts of the work are original and informative.

Weakness:
Some of the reviewers complained that the experimental evaluation was not thorough enough. This is partially something that the entire field of MARL suffers from but the authors should have at least compared to MAPPO in their original submission, rather than doing this only during the rebuttals.
The method it also rather complicated and it would be great to have a more in-depth ablation study of the different components.

On balance I believe it would be preferable to submit an updated manuscript which addresses all of the issues raised by the reviewers to a future conference for a fresh and unbiased set of reviews.